# Genome-Wide Analysis of Amino Acid Transporter Gene Family Revealed That the Allele Unique to the Aus Variety Is Associated with *Amino Acid Permease 17* (*OsAAP17*) Amplifies Both the Tiller Count and Yield in Indica Rice (*Oryza sativa* L.)

Itishree Nayak [1,2], Bijayalaxmi Sahoo [1], Chinmay Pradhan [2], Cayalvizhi Balasubramaniasai [1], Seenichamy Rathinam Prabhukarthikeyan [3], Jawahar Lal Katara [1], Jitendriya Meher [1], Sang-Min Chung [4], Abdel-Rhman Z. Gaafar [5], Mohamed S. Hodhod [6], Bhagwat Singh Kherawat [7], Chidambaranathan Parameswaran [1,*], Mahipal Singh Kesawat [8,*] and Sanghamitra Samantaray [1]

[1] Crop Improvement Division, ICAR-National Rice Research Institute, Cuttack 753006, India; itishreenayak2616@gmail.com (I.N.); bijyalaxmisahoo7@gmail.com (B.S.); cayalshiv@gmail.com (C.B.); jawaharbt@gmail.com (J.L.K.); jmehercrri@gmail.com (J.M.); smitraray@gmail.com (S.S.)

[2] Department of Botany, Utkal University, Vanivihar, Bhubaneswar 751004, India; cpradhan.bot@utkaluniversity.ac.in

[3] Crop Protection Division, ICAR-National Rice Research Institute, Cuttack 753006, India; prabhukarthipat@gmail.com

[4] Department of Life Science, Dongguk University, Goyang 10326, Republic of Korea; smchung@dongguk.edu

[5] Department of Botany and Microbiology, College of Science, King Saud University, Riyadh 11451, Saudi Arabia; agaafar@ksu.edu.sa

[6] Faculty of Biotechnology, October University for Modern Sciences & Arts, 6th of October City 12566, Egypt; mshodhod@msa.edu.eg

[7] Krishi Vigyan Kendra, Bikaner II, Swami Keshwanand Rajasthan Agricultural University, Bikaner 334603, India; skherawat@gmail.com

[8] Department of Genetics and Plant Breeding, Faculty of Agriculture, Sri Sri University, Cuttack 754006, India

* Correspondence: agriparames07@gmail.com (C.P.); mahibiotech@snu.ac.kr (M.S.K.)

**Abstract:** Amino acid transporters (AATs) play a crucial role in facilitating the movement of amino acids across cellular membranes, which is vital for the growth and development of plants. Amino acid permease (AAP), which belongs to the AAT family, has been the subject of extensive functional research in plants. Although its importance is recognized, a comprehensive grasp of this family's dynamics in indica rice remains lacking. In this investigation, a total of 27 AAP genes were identified in the genome of indica rice. Further, the phylogenetic analysis unveiled that the 69 AAP genes from both the model species and other plant species could be classified into 16 distinct subfamilies. The analysis of chromosomal mapping revealed an uneven distribution of the 27 *OsAAP* genes across the 12 rice chromosomes. Notably, the *OsAAP* family displayed a total of 10 duplicated gene pairs, along with the identification of numerous conserved motifs. The examination of cis-elements within OsAAP genes unveiled that their promoters contain cis-elements related to phytohormones, plant growth and development, as well as stress responses. Additionally, transcriptome profiling demonstrated that a substantial portion of these genes exhibited responsiveness to various hormones, with their activation spanning multiple tissues and developmental stages in rice. The study identified miRNAs with a specific affinity for *OsAAP* genes. Out of the 27 *OsAAP* genes investigated, seventeen were discovered to be targeted by a total of forty-three miRNAs. Furthermore, the aus allele of *OsAAP3* that we named *OsAAP17* was validated for its effect on productive tillers and yield, and seventeen genetic variants of *OsAAP17* were found to be associated with a culm number in indica rice. In addition, indica rice varieties were monomorphic, while aus genotypes displayed polymorphism for *OsAAP17* gene-specific in/dels. Moreover, in Season II (rabi season), it was found that the aus allele of *OsAAP17* increased the number of productive tillers and the single plant yield by 22.55% and 9.67%, respectively, in a recombinant inbred population created by crossing N22 and JR 201. Remarkably, this enhancement was more pronounced during the dry cultivation season, highlighting the influence of environmental factors in the regulation of tiller numbers mediated by *OsAAP17*. The

discoveries presented here lay a strong foundation for further exploration into the roles of *OsAAP* family genes across a range of developmental processes. Therefore, the identified allelic variations in the utilization of *OsAAP17* has the potential to enhance rice crop production via molecular breeding in the changing climate scenario.

**Keywords:** rice; productive tillers; yield; aus; amino acid permease; introns

## 1. Introduction

Nitrogen plays a crucial role in the growth and development of plants. The nitrogen assimilation pathway, which serves as the main mechanism for transporting organic nitrogen within plant roots, transforms inorganic nitrogen compounds like nitrate and ammonium into amino acids [1–4]. Additionally, amino acids are transported through both the xylem and phloem, moving from the source organs to the sink organs [5,6]. According to Tegeder and Rentsch [7], higher plants have the capability to directly uptake amino acids from the soil. Amino acids serve as the fundamental constituents of proteins and enzymes, both of which are essential for shaping plant architecture and facilitating metabolic processes. Further, these amino acids play a vital role in supplying the precursors for diverse components such as nucleotides, chlorophyll, secondary metabolites, plant hormones, and lignins [2,3,8]. In plants, amino acids are moved from the roots to other cellular locations via specialized proteins called amino acid transporters (AATs) [4]. Amino acid permease (AAP), which belongs to the AAT family, has been the subject of extensive functional research in plants. AAP gene family members have been identified in various plant species, including Arabidopsis [3], rice [9], poplar [10], *Solanum tuberosum*, and *Glycine max* [11]. Recently, the AAP gene family has garnered significant attention from plant scientists, emerging as a crucial area of focus. These transporters fulfill various functions, including involvement in processes like seed filling, the transfer of amino acids between the xylem and phloem, loading amino acids into the phloem, and absorbing amino acids from the soil into plants [7]. The Arabidopsis genome encodes eight AAP transporters (*AtAAP1-AtAAP8*), which play a crucial role in transporting diverse amino acids to facilitate the organic nitrogen utilization within both source and sink organs. For instance, the role of *AtAAP1* has been substantiated in importing neutral, uncharged amino acids into root cells and developing embryos, thereby playing a substantial part in the synthesis of storage proteins and influencing the seed yield in Arabidopsis [12–14]. *AtAAP2* is responsible for transporting Glu and neutral amino acids, serving as a crucial component for amino acid movement from the xylem to the phloem [15]. Further, *AtAAP3* participates in the absorption of both neutral and basic amino acids [16]. *AtAAP4* is accountable for importing amino acids Proline and Valine, which are electrically neutral [1], while the broad-specificity *AtAAP5* facilitates the transport of anionic, neutral, and cationic amino acids [17]. *AtAAP6* influences the contents of Phe, Lys, Asp, and Leu in sieve elements, governing rosette width and seed volume in Arabidopsis [18]. *AtAAP8*, identified as a high-affinity transporter for acidic amino acids, plays a crucial role in seed development and yield [16,19]. *VfAAP1* and *VfAAP3* transport a wide array of amino acids, with *VfAAP1* displaying a preference for Cys and *VfAAP3* for Lys and Arg in *Vicia faba* [20]. *StAAP1* is predominantly expressed in leaves, and its suppression via antisense technology leads to decreased amino acid content in transgenic *Solanum tuberosum* [21]. *PvAAP1*, expressed in xylem parenchyma cells, epidermal cells, and the phloem, plays a crucial role in the transfer between the xylem and phloem, as well as in phloem loading, thus enabling the transportation of amino acids to sink tissues in *Phaseolus vulgaris* [22]. Additionally, *PtAAP11* has been proposed to play a significant role in xylem formation by supplying Pro in *Populus trichocarpa* [23]. Recent findings demonstrate that the ectopic expression of *PsAAP1* positively affects the transport of amino acids from the source to the sink organs, consequently influencing plant NUE in *Pisum sativum* [24], while *PsAAP6* is involved in

nodule nitrogen metabolism, export, and overall plant nutrition [25]. Furthermore, due to the overexpression of *OsAAP4* regulating the allocation of neutral amino acids via its two splicing variants, it enhances rice tillering and the grain yield [26]. Until now, a total of 19 AAP transporters have been identified and characterized in japonica rice, among which *OsAAP3* stands out for its role in transporting two primary basic amino acids, lysine and arginine. Furthermore, *OsAAP3* is also responsible for the transportation of amino acids such as Serine, Leucine, Methionine, Lysine, Arginine, Histidine, Glutamine, Alanine, and Glycine [27].

The productive tillers are crucial traits that significantly impact the rice yield and the overall productivity. In the context of rice, multiple genes have been studied and identified for their role in regulating the number of productive tillers. For example, rice nitrate transporter OsNPF7.2 [28], Rice MONOCULM 1 or *OsMOC1* [29], *OsSPL14* (Os SQUAMOSA PROMOTER BINDING PROTEIN LIKE gene) [30], *OsCKX2* (CYTOKININ OXIDASE2) [31], PROG1(PROSTRATE GROWTH1) [32], NIN-Like Protein OsNLP4 [33], HTD2 (HIGH TILLERING AND DWARF 2) [34], DWARF27 (D27) [35], OsBRXL4 (BREVIS RADIX LIKE 4) [36], OsMPH1 [37], HTD1 (HIGH TILLERING DWARF 1) [38], OsMADS57 [39], TAC1 [40], DWARF3 [41], OsPIN1 [42], and OsmiR393 [43]. A recent report has highlighted that, by inhibiting the expression of the *OsAAP3* gene, there was an observed increase in yield attributed to the regulation of the concentrations of two fundamental amino acids [27]. Further, *OsAAP3* stimulates bud outgrowth and contributes to augmenting the tiller count in japonica rice varieties, thereby enhancing the overall rice productivity. *OsAAP3* plays a pivotal role in regulating nitrogen utilization efficiency (NUE) in rice. Furthermore, a low *OsAAP3* gene expression in indica rice genotypes was found to be correlated with an increase in effective tillers compared to japonica genotypes [27]. The gene *OsAAP17* (BGIOSGA023082) in indica rice is a counterpart to *OsAAP3* (LOC_Os06g36180) in japonica rice. However, the effect of the aus allele of *OsAAP17* gene is not fully understood. The aus ecotypes of rice are recognized as repositories of genes/QTLs associated with abiotic stress tolerance. Notably, from these aus ecotypes, genes like Pstol1 are used for low phosphorus tolerance [44], qDTYs for drought tolerance during the reproductive stage [45], and qHTSF for heat tolerance [46]. In addition, aus ecotypes demonstrate greater resilience to yield reductions triggered by climate change in comparison to other ecotypes [47]. The discovery of novel genes and alleles within aus ecotypes offers valuable insights into the mechanisms of evolutionary adaptation [48]. Though specific QTLs/genes from aus ecotypes have been pinpointed and incorporated into breeding programs to enhance abiotic tolerance and climate resilience, the exploration of novel genes/alleles within aus ecotypes pertaining to the traits related to yield has not been extensively investigated.

With the recent strides made in DNA sequencing technology, there has been an exponential increase in the sequencing of plant genomes. While genome sequence databases have furnished researchers with a trove of encoded data, the genes unearthed within these plant species' genomes remain largely uncharacterized, especially concerning their functions and regulatory mechanisms. The task of conducting structural and functional analyses of these genes has become an intricate challenge [49–51]. However, a comprehensive exploration of the entire AAP family in indica rice remains unexplored. Rice occupies a significant place as both a crucial cereal crop and a model plant in scientific investigation [51,52]. The main objective of this study was to pinpoint and characterize the AAP family in indica rice. In this study, we undertook an exhaustive genome-wide examination of the AAP gene family in rice, utilizing an array of computational tools. Further, we delved into the physical and biochemical attributes, gene architecture, motif and chromosomal arrangement, instances of gene duplication, three-dimensional structure, protein–protein interactions, and plausible miRNA candidates targeting *OsAAP*. In addition, to obtain insights into their functional roles, we evaluated the expression of the *OsAAP* genes across various tissues and after phytohormone treatments. Several *OsAAP* genes displayed a significant expression in diverse tissues and stages of development, presenting themselves

as promising candidates for subsequent investigation and potential utilization in improving the rice yield and stress resilience. Previously, it was discovered that the indica allele of the *OsAAP3* gene outperformed the japonica allele in terms of the tiller number and yield-related traits. Furthermore, the aus allele of *OsAAP3* that we named *OsAAP17* was validated for its effect on productive tillers and yield and seventeen genetic variants of *OsAAP17* were found to be associated with the culm number in indica rice. In addition, the rice varieties were monomorphic, while the aus genotypes displayed polymorphism for *OsAAP17* gene-specific in/dels. In addition, within the UPGMA phylogeny, three major clusters were identified, two of which were aus-specific clusters. Moreover, the aus allele of *OsAAP17* increased the number of productive tillers and the single plant yield by 22.7% and 9.6%, respectively, in a recombinant inbred population created by crossing N22 and JR 201. Notably, the increase in the number of productive tillers and yield was relatively higher during the dry season of cultivation, indicating that environmental effects on *OsAAP17*-mediated tiller number regulation were present. The discoveries presented here lay a strong foundation for further exploration into the roles of *OsAAP* family genes in various developmental processes. Additionally, these findings provide supporting evidence that genetic variations within the *OsAAP17* gene could have an impact on the traits associated with the rice yield. Therefore, the identified allelic variations in *OsAAP17* present an opportunity to improve the rice yields via molecular breeding in the changing climate scenario.

## 2. Materials and Methods

### 2.1. Identification and Characterization of Members of the OsAAP Gene Family within the Indica Rice Genome

The Pfam database [53] was utilized to obtain a hidden Markov Model (HMM) profile representing the conserved domain (PF01490) of OsAAP. This profile was employed in a search across the rice genome database to identify OsAAP family members. Further identification was conducted using the NCBI-CDD [54] and SMART databases [55]. For each OsAAP protein, the theoretical isoelectric point (PI) and molecular weight (MW) were analyzed using an isoelectric point calculator [56]. In addition, the subcellular localization of the encoded OsAAP proteins was predicted using PSORT and BUSCA tools [57,58].

### 2.2. Phylogenetic Tree, Syntenic Relationships, and Gene Structure Analysis

Arabidopsis OsAAP protein sequences were acquired from Ensembl Plants (https://plants.ensembl.org/index.html) (accessed on 10 August 2023). Subsequently, phylogenetic trees were constructed using MEGA 11 [59]. In this process, sequence alignment was carried out using ClustalW, while the Maximum Likelihood method was applied for building the phylogenetic tree. The tree's reliability was assessed using the bootstrap method with 1000 replicates. A comparative genomic synteny analysis was executed to ascertain the connections among *OsAAP*, across various species including *Oryza sativa Japonica*, *Oryza rufipogon*, *Oryza nivara*, and *Hordeum vulgare*. This analysis was carried out utilizing the circoletto program, a genome visualization tool tailored for visualizing genomic relationships. To visualize the exon–intron structure of *OsAAP* genes, GSDS [60] was employed.

### 2.3. Localization of Chromosomes, Duplication of Genes, Cis-Regulatory Elements, and Gene Ontology Analysis

Chromosome localization data for the OsAAP genes was acquired from Ensembl Plants (http://plants.ensembl.org/biomart/martview) (accessed on 10 August 2023), aiding in the mapping of their positions on the chromosomes. The mapping of the *OsAAP* gene family members was accomplished using PhenoGram v1 [61]. For gene duplication and the Ka/Ks value analysis, McScan tools v1 [62] and TBtools v.1 [63] were employed. To analyze the promoter elements, the 2000 bp sequence upstream of the *OsAAP* genes was subjected to assessment via the PlantCARE webserver [64]. The GO enrichment analysis of OsAAP proteins was carried out using agriGO [65].

### 2.4. Conserved Motifs, Transmembrane Domains, and the 3D Structure

Furthermore, conserved motifs within the OsAAP protein sequences were delineated using the MEME webserver [66]. To analyze the transmembrane helix and topology of the OsAAPs, the SOSUI tool (http://www.cbs.dtu.dk, http://harrier.nagahama-i-bio.ac.jp) (accessed on 15 August 2023) and TMHMM tool (http://www.cbs.dtu.dk/services/TMHMM/) (accessed on 15 August 2023) were employed. For the creation of the three-dimensional structure of OsAAP, the Phyre2 web server was utilized [67].

### 2.5. Gene Expression, miRNA, and the Analysis of Protein–Protein Network

The expression values specific to various tissues for the 27 *OsAAP* genes were gathered from the RiceXPro database (http://ricexpro.dna.affrc.go.jp) (accessed on 18 August 2023). Heatmaps visualizing this data were generated using ClustVis [68]. The coding sequence of *OsAAP* was employed to identify potential miRNAs that could target *OsAAP*. This identification process was conducted using the psRNATarget database (https://www.zhaolab.org/psRNATarget/) (accessed on 18 August 2023) with its default parameters. The interaction network of OsAAP proteins was investigated utilizing the STRING online server (https://string-db.org/cgi) (accessed on 20 August 2023).

### 2.6. Primer Development of OsAAP17 Gene (LOC_Os06g36180)

The AAP3 gene inserts and deletions, including the promoter region, were obtained from the rice SNP Seek database (https://snp-seek.irri.org/) (accessed on 22 August 2023) [69]. Furthermore, the annotated AAP3 gene sequence of Nipponbare was obtained from the RGAP-Rice Genome Annotation Project database (http://rice.plantbiology.msu.edu/) (accessed on 23 August 2023) and used to develop in/del primers (Table S10) with the IDT primer quest tool (https://www.idtdna.com/pages/tools/primerquest) (accessed on 25 August 2023). Only in/dels with nucleotide sizes greater than 10 bp were considered for the primer design. Global polymorphic variant In/dels (GPV-In/dels) is the name given to the newly developed marker. The *OsAAP17* primers that were created were then evaluated for gene diversity and gene allelic effects.

### 2.7. OsAAP17 Gene Diversity Analysis in Rice Varieties and Aus Genotypes

The seeds of 131 different indica rice varieties and 24 different aus genotypes (BAAP population) were obtained from the ICAR-NRRI Cuttack Gene Bank. The genotypes used in the analysis are listed in Table S11a,b. The seeds were kept at room temperature for a day before being dried at 40 °C for three days to ensure uniform germination. The seeds were line-sown in soil for DNA isolation, and 15-day-old green leaves were collected and immediately frozen in liquid nitrogen before being stored at −80 °C. The genomic DNA from these young leaves was isolated using the cetyltrimethyl ammonium bromide (CTAB) method [70]. To quantify the isolated DNA, agar gel electrophoresis (0.8%) was used. The isolated genomic DNA was diluted (50 ng/L) in nuclease-free water for PCR amplification. The PCR reaction mixture included $1\times$ Taq buffer (10 mMTris-HCl, 50 mMKCl, pH 8.3), 0.2 lM forward and reverse in/del primers, 1.5 mM $MgCl_2$, 0.2 lM dNTPs, 25 ng template DNA, and 0.2 U Taq DNA polymerase enzyme (GeneiTaq, Bangalore, India). In an Eppendorf PCR cycler, the initial denaturation was set to 95 °C for 3 min, followed by 35 cycles of 94 °C for 1 min, primer annealing for 1 min at 65 °C, elongation for 45 s at 72 °C, and a final extension at 72 °C for 10 min. The amplified products were run in 4% agarose gel, and the results were documented using the Biorad UV gel documentation system. Furthermore, the genotypes' alleles were manually scored based on amplicon length. Allelic data were used for gene diversity and the neighbor-joining phylogenetic tree construction using Power Maker software 3.25 [71].

### 2.8. Development of Mapping Population between N22 and JR201

Based on our previous study on spikelet fertility in the field conditions [72], the aus ecotype, N22, a drought and high-temperature tolerant variety, and JR201, a variety suited

for rainfed and upland cultivation, were chosen for the development of the mapping population. Furthermore, one of the five in/dels primers, *OsAAP17*-5, revealed polymorphism between the two parents, making it suitable for validating the *OsAAP17* gene alleles on yield-related traits in the mapping population. The work on developing a mapping population between N22 and JR201 began in 2016–17, and the single seed descent method was used to develop the mapping population of recombinant inbred lines, as previously reported by Kanbar et al. [73]. The F6 lines' seeds were planted during the Kharif (August–December) and rabi (January–May) seasons of 2020–21 and used to evaluate yield-related traits. Table S13 contains the weather data for the growing season (Table S13).

*2.9. Screening of Mapping Population for Tiller Number and Yield-Related Traits*

Approximately 100 seeds of parents and individual lines of F6 mapping populations from N22 to JR201 were sown in the nursery bed and line transplanted in the field for yield-related trait evaluation. For the study, two replications per genotype were maintained, with each replication consisting of 30 seedlings transplanted in two different lines in the field. The genotypes were line transplanted and the phenotypes were evaluated using the alpha lattice design, as previously reported [72]. Six plants per genotype were randomly selected at physiological maturity, i.e., three individual plants per replication, and used to collect data on various yield-related traits. The topmost panicle's panicle length (cm) and number of productive tillers (nos.) were collected from the field. Furthermore, at the maturity stage, six plants per genotype were harvested, and traits such as the total spikelet number (nos.), filled grain (nos.), unfilled grain (nos.), and single plant yield (g) were measured for further analysis. The panicle length was measured in centimeters, from the base of the panicle to the topmost spikelet. The total spikelet number, filled grain number, and unfilled grain number were manually counted for three panicles per plant and three plants per replication. Furthermore, the single plant yield from three plants per replicate was measured.

*2.10. Linear Regression and Classification Analysis*

The SNPs and in/dels found in the *OsAAP17* gene from the SNP seek database were also investigated for their relationship with the culm number using linear regression, lasso, and random forest models [74]. Using *OsAAP17* SNPs and In/Dels, a random forest classifier was used to classify the rice genotypes into different ecotypes. This analysis was carried out in the same manner as previously reported by Vejchasarn et al. [75], and *OsAAP17* genetic variants and genotype ecotype information from the rice SNP seek database were used to assess the classification error using random forest classifier models. Furthermore, the random forest classifier's varImp function was used to assess variant importance. Furthermore, using the culm number as a response variable, the LASSO model was used for genetic variant selection [76].

*2.11. Statistical Analysis*

The data analysis tool in MS Excel 2016 was used to determine descriptive statistics such as the mean, standard error, median, mode, standard variation, sample variance, kurtosis, skewness, and range. Individual lines in the mapping population were grouped based on allelic information, and mean differences for both seasons were tested for statistical significance using the Z-test at 5% and 1% significance levels. Similarly, the Z test for mean difference was computed for various traits across two seasons.

**3. Results**

*3.1. Identification and Characterization of OsAAP Genes across the Entire Genome of Indica Rice and Evolutionary Analysis*

This research uncovered 27 AAP genes within rice (*Oryza sativa* indica). Among these, all 27 *OsAAP* genes were projected to produce proteins ranging in length from 148 to 487 amino acids, as shown in Tables 1 and S1.



**Table 1.** The nomenclature and attributes of the potential amino acid transporter gene family (*OsAAP*) proteins in indica rice.

| Proposed Gene Name | Gene ID | Genomic Location | Orientation | CDS Length (bp) | Protein Length (aa) | Molecular Weight (KDa) | Isoelectric Point (pI) | GRAVY | Predicted Subcellular Localization |
|---|---|---|---|---|---|---|---|---|---|
| *OsAAP1* | BGIOSGA001188 | 1:27819467–27819727 | Reverse | 1365 | 454 | 48.57 | 8.26 | 0.72 | plasma membrane |
| *OsAAP2* | BGIOSGA000460 | 1:40537731–40537886 | Reverse | 1566 | 521 | 56.07 | 8.15 | 0.63 | plasma membrane2 |
| *OsAAP3* | BGIOSGA000395 | 1:41725345–41726478 | Reverse | 1398 | 465 | 49.94 | 7.72 | 0.54 | plasma membrane3 |
| *OsAAP4* | BGIOSGA000394 | 1:41739541–41742697 | Reverse | 1401 | 466 | 50.02 | 7.42 | 0.62 | plasma membrane4 |
| *OsAAP5* | BGIOSGA000378 | 1:41942366–41943340 | Reverse | 1467 | 488 | 52.87 | 7.65 | 0.5 | plasma membrane |
| *OsAAP6* | BGIOSGA007327 | 2:125686–126493 | Forward | 1455 | 484 | 53.22 | 6.64 | 0.37 | plasma membrane |
| *OsAAP7* | BGIOSGA008784 | 2:29016053–29016202 | Forward | 1269 | 422 | 44.84 | 7.2 | 0.8 | plasma membrane |
| *OsAAP8* | BGIOSGA008950 | 2:31930792–31931241 | Forward | 1329 | 442 | 47.07 | 5.72 | 0.42 | plasma membrane |
| *OsAAP9* | BGIOSGA014910 | 4:21924811–21927625 | Reverse | 447 | 148 | 15.31 | 7.76 | 0.44 | organelle membrane |
| *OsAAP10* | BGIOSGA016659 | 4:22997818–22998441 | Forward | 1236 | 411 | 43.11 | 7.67 | 0.56 | plasma membrane |
| *OsAAP11* | BGIOSGA016661 | 4:23016996–23017175 | Forward | 456 | 151 | 15.89 | 6.97 | 0.45 | endomembrane system |
| *OsAAP12* | BGIOSGA014538 | 4:27155165–27155314 | Reverse | 1278 | 425 | 44.76 | 7.78 | 0.8 | plasma membrane |
| *OsAAP13* | BGIOSGA017288 | 4:32739933–32740566 | Forward | 1413 | 470 | 51.4 | 8.31 | 0.44 | plasma membrane |
| *OsAAP14* | BGIOSGA018471 | 5:10135284–10135993 | Reverse | 1371 | 456 | 49.94 | 7.13 | 0.54 | plasma membrane |
| *OsAAP15* | BGIOSGA019901 | 5:22018146–22018484 | Forward | 1491 | 496 | 53.51 | 8.73 | 0.54 | plasma membrane |
| *OsAAP16* | BGIOSGA020459 | 5:30590867–30591034 | Forward | 1446 | 481 | 51.04 | 7.06 | 0.72 | chloroplast outer membrane |
| *OsAAP17* | BGIOSGA023082 | 6:21947609–21953508 | Forward | 1464 | 487 | 52.78 | 7.42 | 0.4 | plasma membrane |
| *OsAAP18* | BGIOSGA023083 | 6:21972799–21972954 | Forward | 1425 | 474 | 50.72 | 7.44 | 0.52 | plasma membrane |
| *OsAAP19* | BGIOSGA021614 | 6:7712971–7714521 | Reverse | 1455 | 484 | 51.9 | 7.74 | 0.46 | endomembrane system |
| *OsAAP20* | BGIOSGA024866 | 7:1787401–1788046 | Reverse | 1458 | 485 | 52.59 | 8.09 | 0.43 | plasma membrane |
| *OsAAP21* | BGIOSGA025496 | 7:9759616–9760347 | Forward | 1356 | 451 | 48.41 | 8.52 | 0.54 | plasma membrane |
| *OsAAP22* | BGIOSGA027915 | 8:1609038–1609402 | Forward | 1344 | 447 | 49.83 | 7.61 | 0.52 | plasma membrane |
| *OsAAP23* | BGIOSGA030830 | 9:14727692–14728921 | Forward | 2310 | 769 | 82.7 | 7.89 | −0.23 | organelle membrane |
| *OsAAP24* | BGIOSGA037101 | 12:4448712–4448945 | Forward | 1449 | 482 | 51.9 | 7.9 | 0.51 | plasma membrane |
| *OsAAP25* | BGIOSGA036516 | 12:4465477–4465710 | Reverse | 1428 | 475 | 51.11 | 7.18 | 0.55 | plasma membrane |
| *OsAAP26* | BGIOSGA036484 | 12:5104622–5104741 | Reverse | 1407 | 468 | 50.01 | 8.29 | 0.52 | plasma membrane |
| *OsAAP27* | BGIOSGA037215 | 12:7432040–7433174 | Forward | 1335 | 444 | 48.83 | 9.08 | 0.59 | plasma membrane |

ID stands for identity; bp represents base pairs; aa denotes amino acids; pI signifies isoelectric point; MW indicates molecular weight; GRAVY stands for Grand average of hydropathy; KDa refers to kilo daltons; and GRAVY represents the grand average of hydropathy.

The protein's molecular weight spanned from 15.31 KDa (OsAAP9) to 82.7 KDa (OsAAP23). The isoelectric point exhibited a range between 5.72 (OsAAP8) and 8.09 (OsAAP27). Among these, 24 proteins encoding OsAAP displayed a pI greater than 7, indicating a prevalence of basic amino acids within most OsAAP proteins. The observed GRAVY values for the 27 OsAAP proteins fell between −0.23 (OsAAP23) and 0.80 (OsAAP12). Moreover, predictions regarding subcellular localization indicated that 22 OsAAP proteins were situated in the plasma membrane, while the rest were distributed across the chloroplast outer membrane (OsAAP16), the endomembrane system (OsAAP11 and OsAAP19), and organelle membranes (OsAAP9 and OsAAP23). Additionally, a correlation between the molecular weight and pI of OsAAP was examined to establish the distribution pattern of OsAAP proteins (Figure S1). These findings unveiled a similarity in molecular weight and the isoelectric point across nearly all OsAAP proteins. Furthermore, an evaluation of the evolutionary connections among *OsAAP* genes encompassed *O. sativa*, *A. thaliana*, *S. bicolor*, *S. moellendorffii*, and *P. patens* AAP genes. To construct the phylogenetic tree for this analysis, MEGA 11 was employed, incorporating 27 AAP proteins from *O. sativa*, 8 AAP proteins from *A. thaliana*, 21 AAP proteins from *S. bicolor*, 8 AAP proteins from *S. moellendorffii*, and 5 AAP proteins from *P. patens* (Figure 1, Table S2).

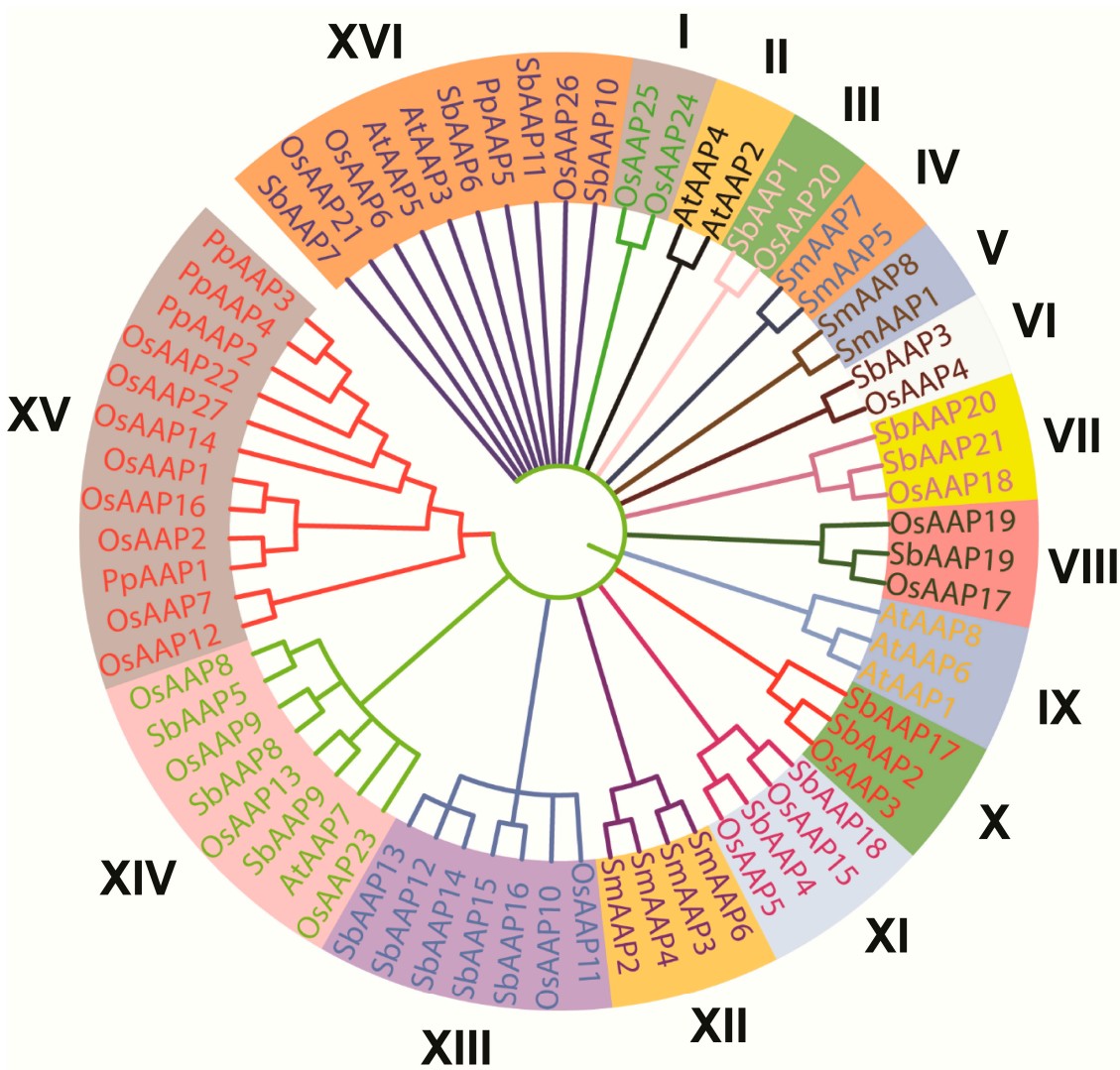

**Figure 1.** The phylogenetic connections between AAP proteins in *A. thaliana*, *O. sativa*, *S. bicolor*, *S. moellendorffii*, and *P. patens* were investigated. Utilizing the neighbor-joining (NJ) technique in MEGAX, a phylogenetic tree was constructed, and bootstrap values were established using 1000 replicates.

The phylogenetic analysis revealed that the 69 AAP proteins from the model and other plant species could be categorized into 16 distinct subfamilies. However, Group XV comprises the highest eight members of the *OsAAP* gene family, while Groups I, II, III, IV, V, VI, VII, VIII, IX, X, XI, XII, XIII, XIV, and XVI contained two, zero, one, zero, zero, one, one, two, zero, one, two, zero, one, four, and three members, respectively (Figure S2).

### 3.2. Analysis of Gene Duplication and Chromosome Distribution of the OsAAP Genes

For a more comprehensive exploration of gene duplication events in rice, we conducted an analysis involving the chromosomal mapping of the identified *OsAAP* family genes. This process resulted in the mapping of 27 *OsAAP* genes onto nine distinct chromosomes (Figure 2). Chromosomes 1 and 4 exhibited the highest count of *OsAAP* genes, with each containing five genes, while chromosome 9 contained the fewest with one gene. The distribution of the 27 *OsAAP* genes across the 12 rice chromosomes was uneven (Figure 2).

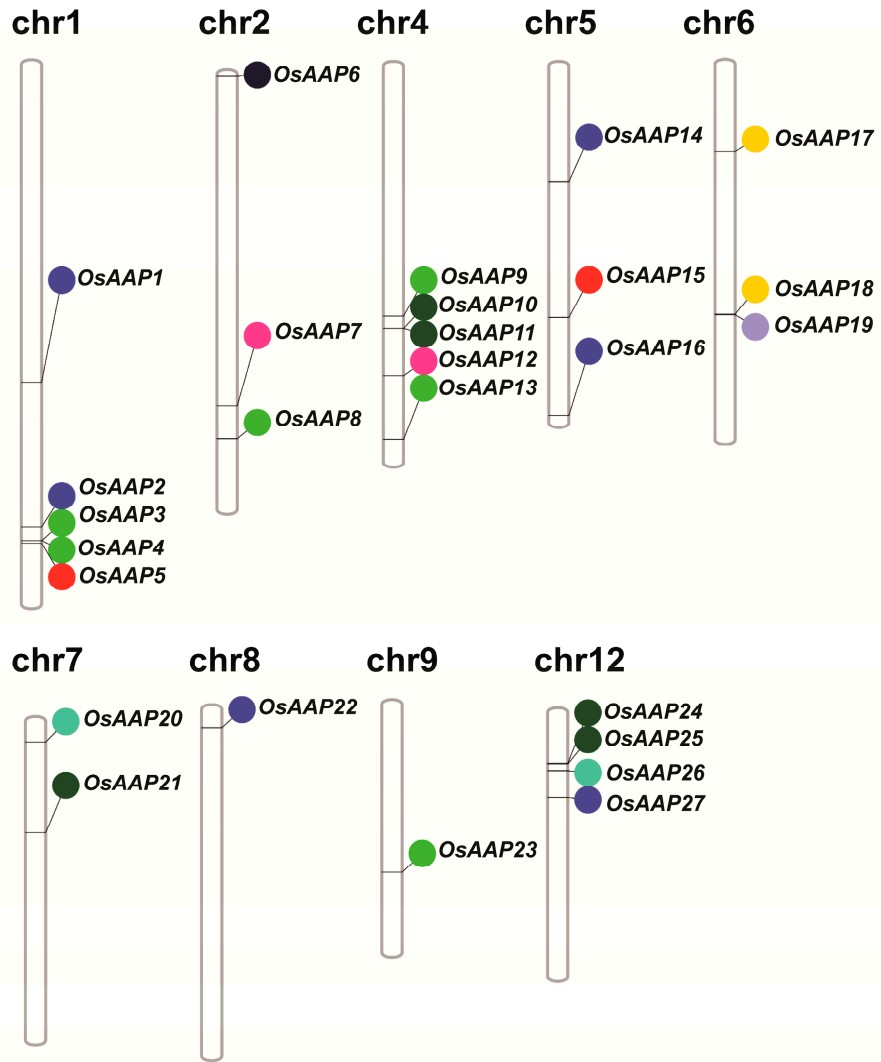

**Figure 2.** Localization of the identified AAP genes on distinct chromosomes within rice. Illustrations in schematic form depicting the dispersion of AAP genes across rice's 12 chromosomes, with gene names positioned on the right-hand side of the chromosomes. Colored circles on individual rice chromosomes denote the precise positions of AAP genes. Chromosome numbers are indicated at the upper section of each chromosome.

Gene duplication is a central driving force in the evolution of gene families. To delve into the evolutionary connections of *OsAAP* genes, this study focused on analyzing the gene duplication events within the *OsAAP* gene group (Figures 3 and S3).

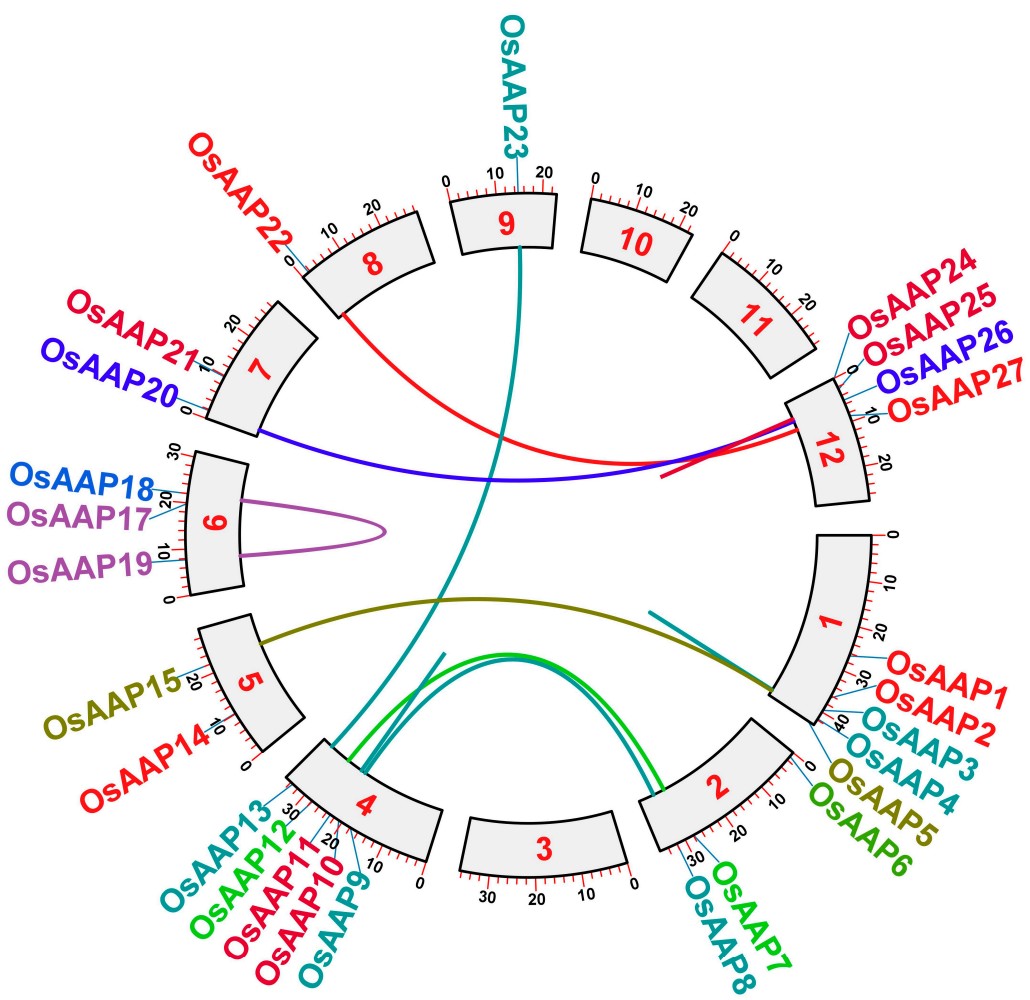

**Figure 3.** The figure, which displays the chromosomal distribution and duplicated AAP gene pairs in rice, features connections between duplicated AAP gene pairs represented by distinct colored lines. This visualization was generated using TB tools.

In total, 10 duplicated gene pairs have been identified (Table S3). The phylogenetic tree of the *OsAAP* genes exposed several instances of gene duplication occurrences (Figure S3). Furthermore, the Ka/Ks scores for these genes were lower than 1, indicating a substantial purifying selection typically occurs, resulting in slight changes or modifications in the duplicated genes. Nevertheless, a solitary gene pair (*OsAAP10-OsAAP11*) displayed a Ka/Ks value exceeding one, suggesting a positive selection effect on this particular gene pair. These insights suggest that the growth of the AAP gene family in rice has predominantly stemmed from both segmental and whole-genome duplications. The genome conservation was visualized via the utilization of the Circoletto Tool (tools.bat.infspire.org/circoletto/) (accessed on 26 August 2023) in the process of conducting the comparative synteny analysis. The investigation into synteny involved the examination of various plant species, including *O. rufipogon*, *O. glaberrima*, *O. nivara*, and *H. vulgare*. The gene sequence of *OsAAP1* exhibited synteny with the gene sequence *ONIVA02G30080* from *O. nivara*. Similarly, the *HORVU.MOREX.r3.2HG0174620* gene from potatoes demonstrated synteny with *OsAAP7* in indica rice. Notably, the gene sequence of *OsAAP* exhibited synteny across these plant

species, as depicted in Figure 4. These findings collectively showed that the *OsAAP* family genes have experienced a conserved evolutionary process.

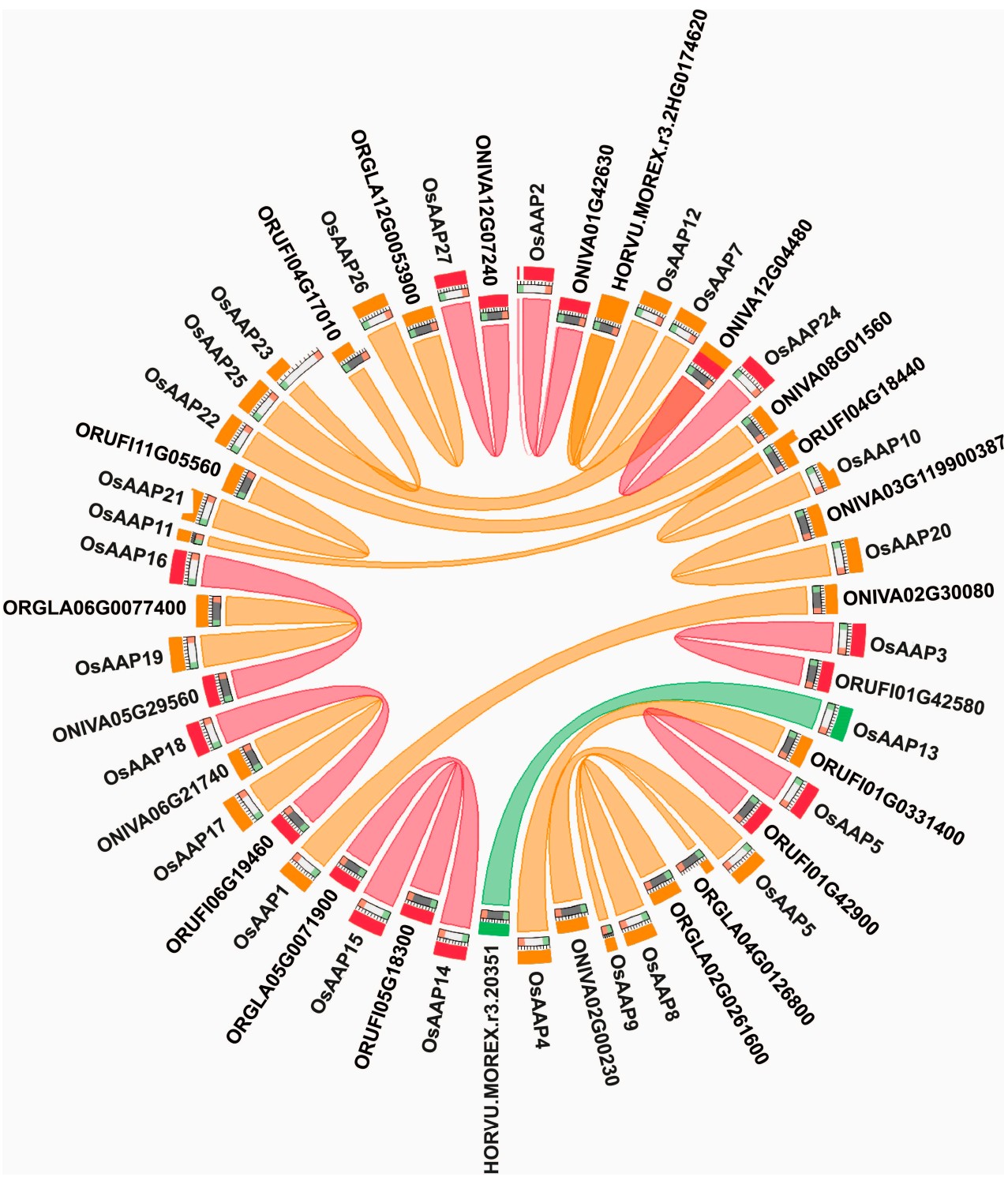

**Figure 4.** Depicting the synteny among *Oryza sativa indica*, *Oryza rufipogon*, *Oryza glaberrima*, *Oryza nivara*, and *Hordeum vulgare*, the aim is to discern the degree of sequence conservation using four distinct colors. The colors red, green, orange, and pink have been designated to indicate varying levels and degrees of evolutionary conservation within the *OsAAP* genes.

### 3.3. Gene Structure and Basic Motif Analysis of OsAAP Gene

The analysis of exon and intron structures was conducted on the genes of the *OsAAP* family to comprehend the diversity in their structural composition and how it contributes to the evolution of these family genes. Based on the gene structure analysis, it was evident that the quantity of introns differed across various subfamilies (Figure 5).

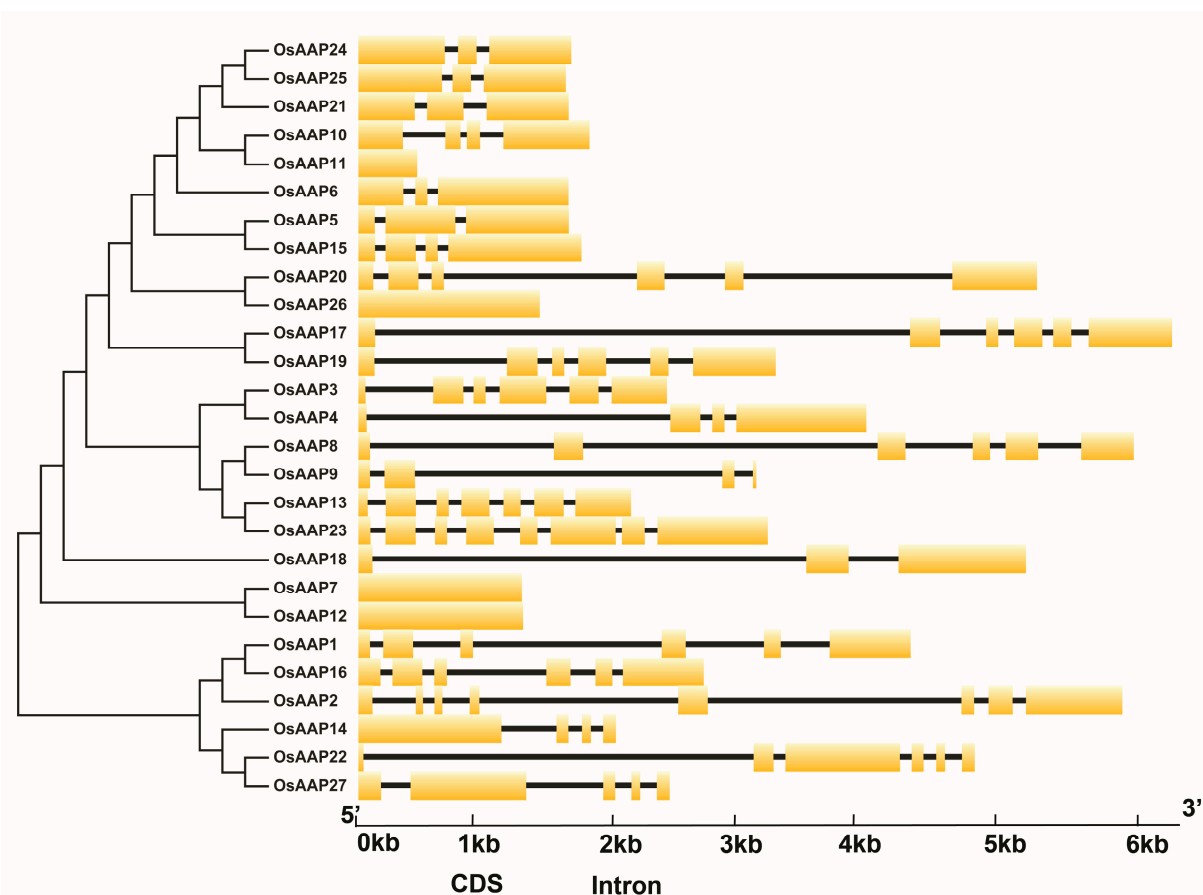

**Figure 5.** Detection of exon–intron arrangement within *OsAAP* genes. Exons are represented by yellow boxes, and introns are indicated by black lines. Box and line lengths are proportionate to gene size. The scale at the bottom can be utilized to estimate exon and intron dimensions.

In the spectrum of AAP subfamily genes, the count of introns spanned from zero to seven; however, the majority exhibited between two to three introns. Notably, *OsAAP23* featured the highest count of seven introns, whereas *OsAAP7*, *OsAAP11*, *OsAAP12*, and *OsAAP26* displayed an absence of introns (Figures 5 and S4). Intriguingly, it was observed that none of the AAP genes possessed 3′- and 5′-untranslated regions (UTRs). Further, for an in-depth exploration of the attributes inherent to OsAAP proteins, we successfully pinpointed 15 conserved motifs within these proteins (Figure 6 and Table S4).

In this investigation, it was found that Motifs 1, 2, 3, 4, and 5 displayed a high level of conservation across most of the OsAAP proteins. In addition, via the alignment of amino acid sequences, it became apparent that OsAAP proteins incorporate these conserved motifs (Figures 7 and S5A). The prediction of both the 3D structure and transmembrane helices of OsAAP proteins was also carried out in this study (Figure S5B,C). The OsAAP protein encompass a range of two to eleven transmembrane helices (Figure S5C). The anticipated transmembrane helices found in OsAAP proteins exhibited a consistent configuration, characterized by a shared amino and carboxy-terminal segment of transmembrane portions, accompanied by a distinct central region. It is worth noting that OsAAP9, OsAAP11, and OsAAP23 deviated from this pattern with a configuration of 2–6 helices and a compact

central loop. Several conserved motifs have been identified within the OsAAP family proteins which potentially play a role in regulating a variety of biological processes.

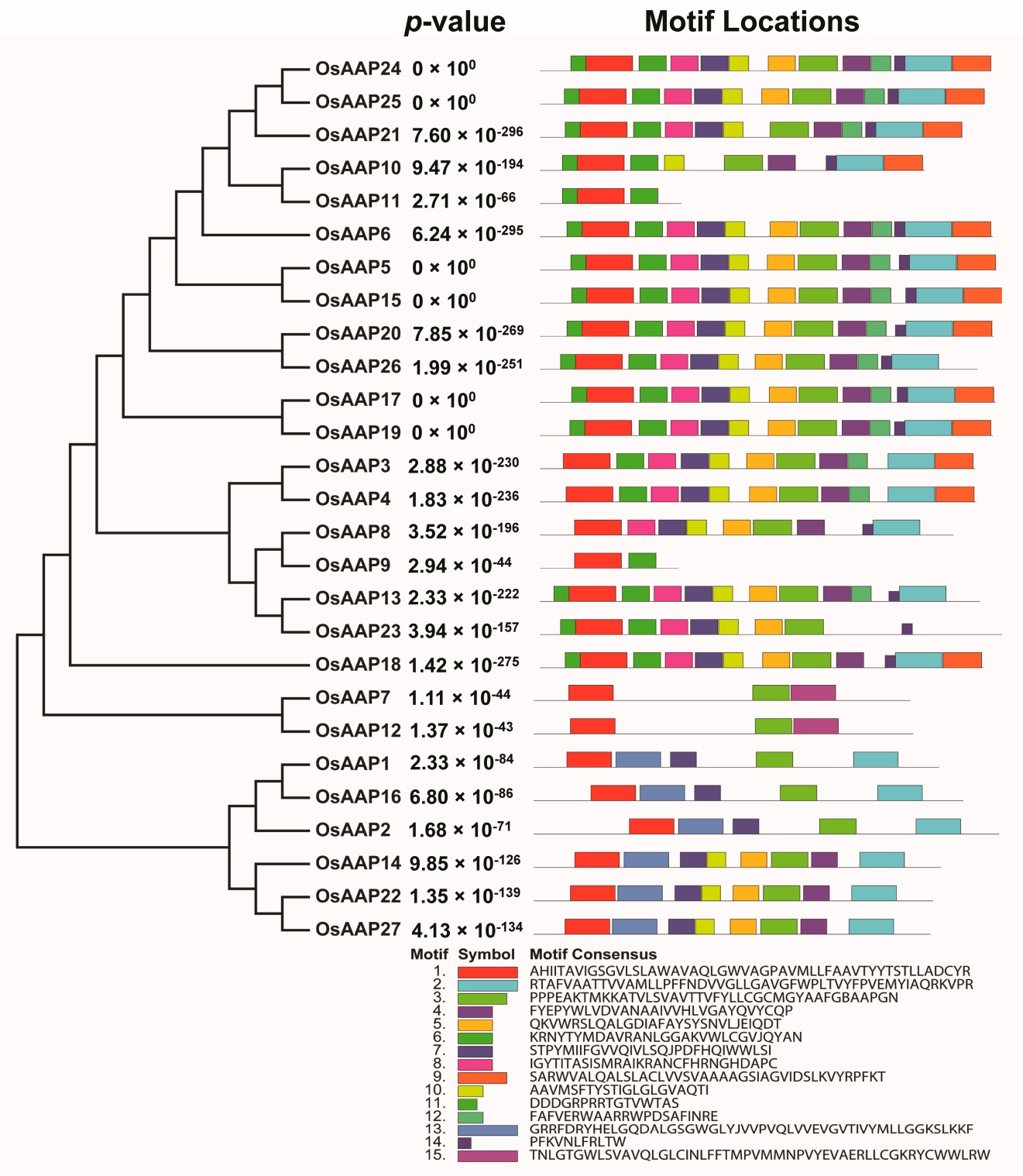

**Figure 6.** Identifying conserved patterns in OsAAP genes led to the discovery of fifteen consistent motifs via the utilization of the MEME database. These distinct conserved motifs, characterized by diverse colored boxes, exhibit varying sizes and sequences.

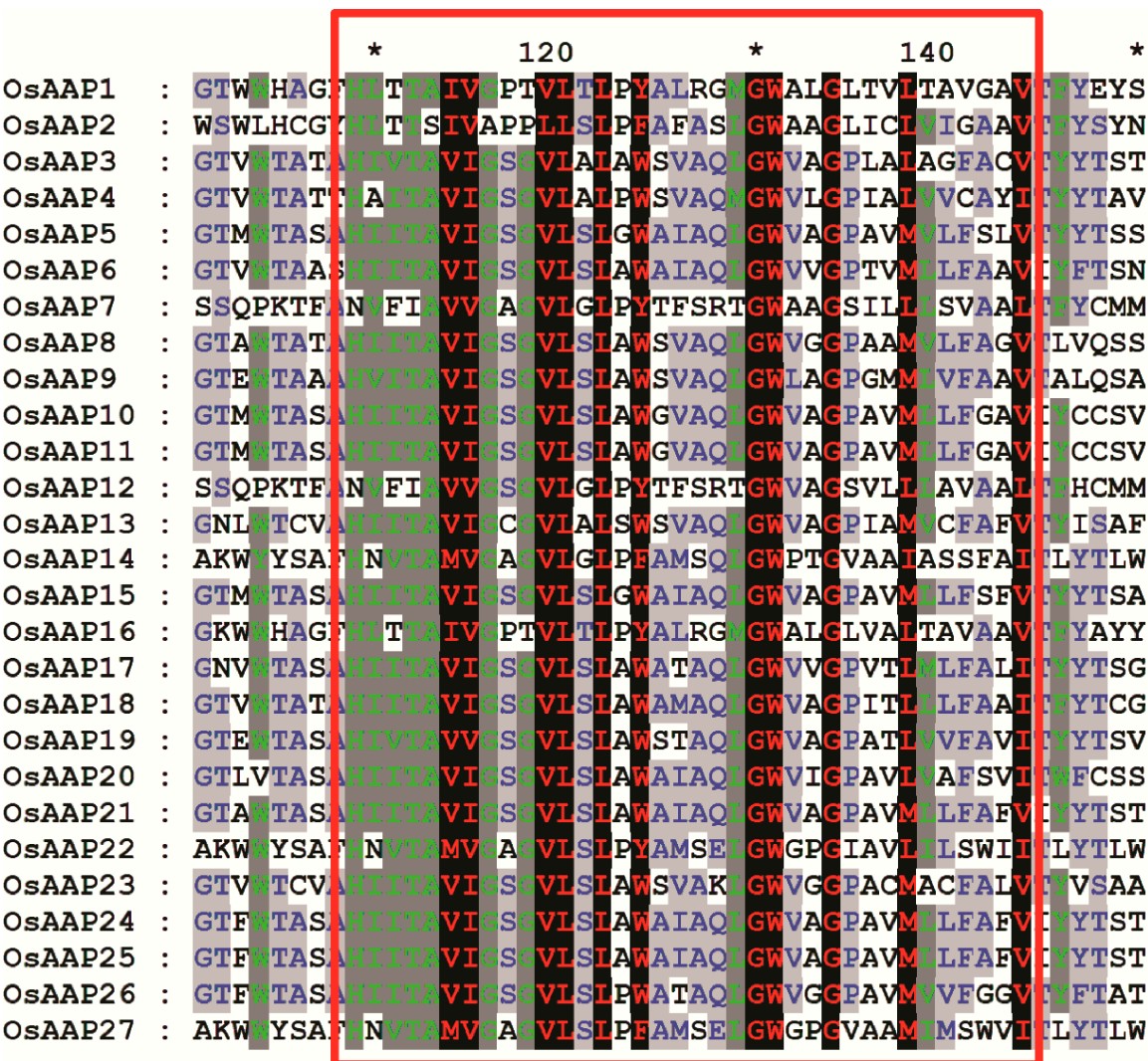

**Figure 7.** The amino acid sequence of the OsAAP protein displays a conserved domain, which is marked by a red box. Amino acids with similar chemical properties are represented with a range of colors and shading. Dashes are used to signify the inclusion of gaps in order to enhance alignment within the homologous region.

### 3.4. Analysis of Promoter Elements and Gene Ontology (GO) Enrichment

To better understand the potential functions of the *OsAAP* family genes, we utilized the PlantCARE online web server to examine the 2000 bp upstream sequences of these genes. This analysis unveiled a multitude of cis-regulatory elements encompassing aspects such as phytohormone responses, light sensitivity, circadian rhythm modulation, seed-specific regulation, and defense and stress responsiveness (Figure 8A,B).

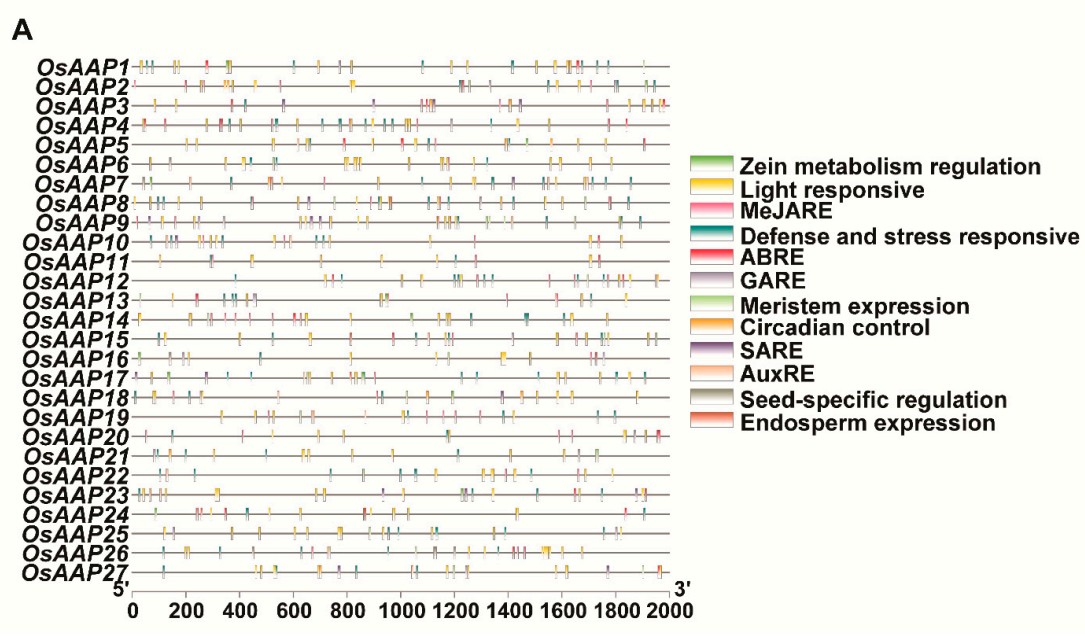

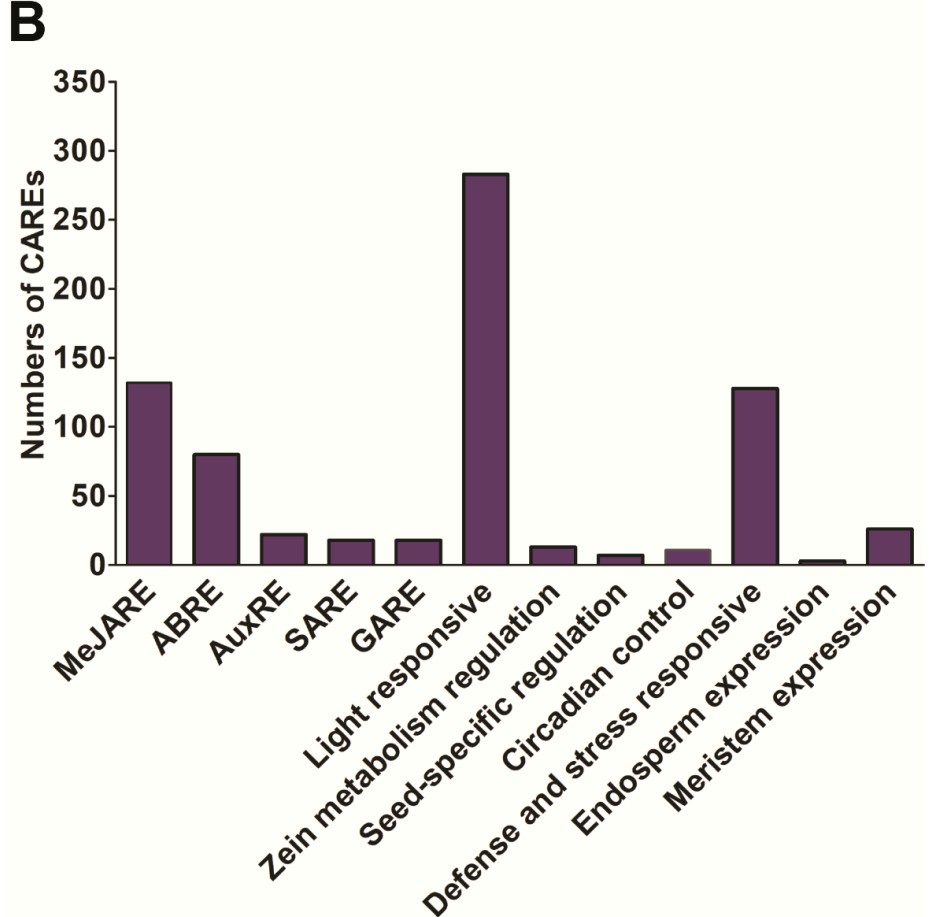

**Figure 8.** CAREs identified in the promoter region of the *OsAAP* gene family. The analysis of these CAREs was conducted using the PlantCARE online server on a 2 kb upstream region. (**A**) The different types of elements present are depicted using distinct colors, including growth and development-related elements, light-responsive elements, stress-responsive elements, and hormone-responsive elements. (**B**) The often encountered CAREs within *OsAAP* promoters.

Regarding hormonal responses, the analysis uncovered the existence of various responsive elements, such as the abscisic acid responsive element (ABRE), different auxin responsive elements (AuxRE), gibberellin responsive elements (GARE), salicylic acid responsive elements (SARE), and methyl jasmonate responsive elements (MeJARE) (Table S5). In the promoter regions of most genes, we identified the presence of MeJARE and ABRE elements. Notably, light responsive, MeJARE, and ABRE elements were prevalent across nearly all *OsAAP* genes. However, the SARE, AuxRE, and GARE-motif elements were only detected in a limited number of promoter regions among the *OsAAP* genes. In stress response, there are elements related to anaerobic induction, drought induction, low temperature response, and salt stress and defense and stress responses. Among these elements, the AU-rich element (ARE) was identified in the promoter regions of 22 genes, while the low-temperature responsiveness (LTR) element was detected in the promoter regions of 19 genes, typically in quantities ranging from 1 to 2 elements. Additionally, TC-rich repeat elements were detected in the promoter regions of 21 genes, typically found in 1–2 copies. Furthermore, MYB binding site (MBS) elements, known for their association with drought response, were also found in the promoter regions of 18 genes (Table S5). These findings indicate that the *OsAAP* family of genes might have a significant function in both plant development and stress response. Furthermore, it is implied that distinctions in promoters could contribute to functional differences within the family, ultimately aiding rice in adapting to diverse abiotic stresses.

To gain a deeper comprehension of the role of *OsAAP* family genes, we performed a Gene Ontology (GO) enrichment analysis. AgriGO effectively annotated and assigned GO terms to all members of the *OsAAP* gene family. This was subsequently corroborated via eggNOG-Mapper, as depicted in Figure S6 and detailed in Tables S6 and S7. Notably, eggNOG-Mapper yielded results consistent with those obtained from AgriGO. In the biological process category (Figure S6A), OsAAP genes displayed enrichment in response to stimulus (GO:00050896), localization (GO:00051179), and transport (GO:0006810). In the cellular category, the enrichment of OsAAP was observed specifically in the membrane (GO:0016020) (Figure S6B). Moreover, within the molecular category, OsAAP genes demonstrated enrichment in transporter activity (GO:0005215) (Figure S6C). Therefore, these findings strongly suggest that OsAAP family genes play a vital role in a wide range of cellular processes in rice.

*3.5. Transcriptome Profiles of OsAAP Family Genes across Distinct Tissues and under Varying Phytohormone Treatments in Rice*

The expression patterns of *OsAAP* family genes were examined in various tissues, developmental stages, and in response to hormone treatments, aiming to elucidate their roles. The expression values for the *OsAAP* gene family were obtained from the Rice Expression Profile Database (RiceXPro) and were subsequently employed to construct heatmaps. Notably, a total of 27 *OsAAP* showed differential expression in different tissues, developmental stages, and hormone treatments, as illustrated in Figure 9A–C.

For example, the expression of *AAP7* and *AAP9* was heightened in various stages: vegetative, reproductive, and ripening leaf blades, as well as in the vegetative and reproductive leaf sheaths and both the vegetative and reproductive roots. Conversely, the induction of an *AAPX18* expression was observed in the anther ovary, embryo, and endosperm. Further, an *AAPX17* expression is prominently stimulated in the anther and endosperm. In contrast, *AAP15* exhibits significant up-regulation in the anther, lemma, and palea. Furthermore, we conducted an analysis of the expression profile of *OsAAP* in both shoots and roots, subjecting them to various plant hormone treatments including gibberellin (GA), cytokinin, auxin, abscisic acid (ABA), brassinosteroid (BRS), and jasmonic acid (JA) (Figure 9B,C). In the shoot, the expression of *OsAAP5* and *OsAAP14* was elevated following both ABA and auxin treatments after 3 h and 6 h. Meanwhile, *OsAAP8* displayed an increase in the expression after 3 h, 6 h, and 12 h of auxin treatment in the shoots. Additionally, the expression of *OsAAP2* and *OsAAP19* increased under the GA treatment after 1 h, 3 h, 6 h,

and 12 h in the shoots (Figure 9B). Further, the expression of *OsAAP8*, *OsAAP15*, and *OsAAP16* exhibited an increase in the root under the JA treatment at 30 min, 1 h, 3 h, and 6 h. Similarly, for *OsAAP15* and *OsAAP16*, the expression levels were elevated in response to the cytokinin treatment at 30 min, 1 h, 3 h, and 6 h in the root. After 3 h and 6 h of auxin treatment in the root, the expression levels of *OsAAP3*, *OsAAP14*, and *OsAAP15* were induced. Following 3 h and 6 h ABA treatments in the root, the transcript levels of *OsAAP9* and *OsAAP11* were observed to increase (Figure 9C). Taken together, these findings indicate that various members of the *OsAAP* gene family could be involved in distinct developmental processes and might respond to phytohormones in rice.

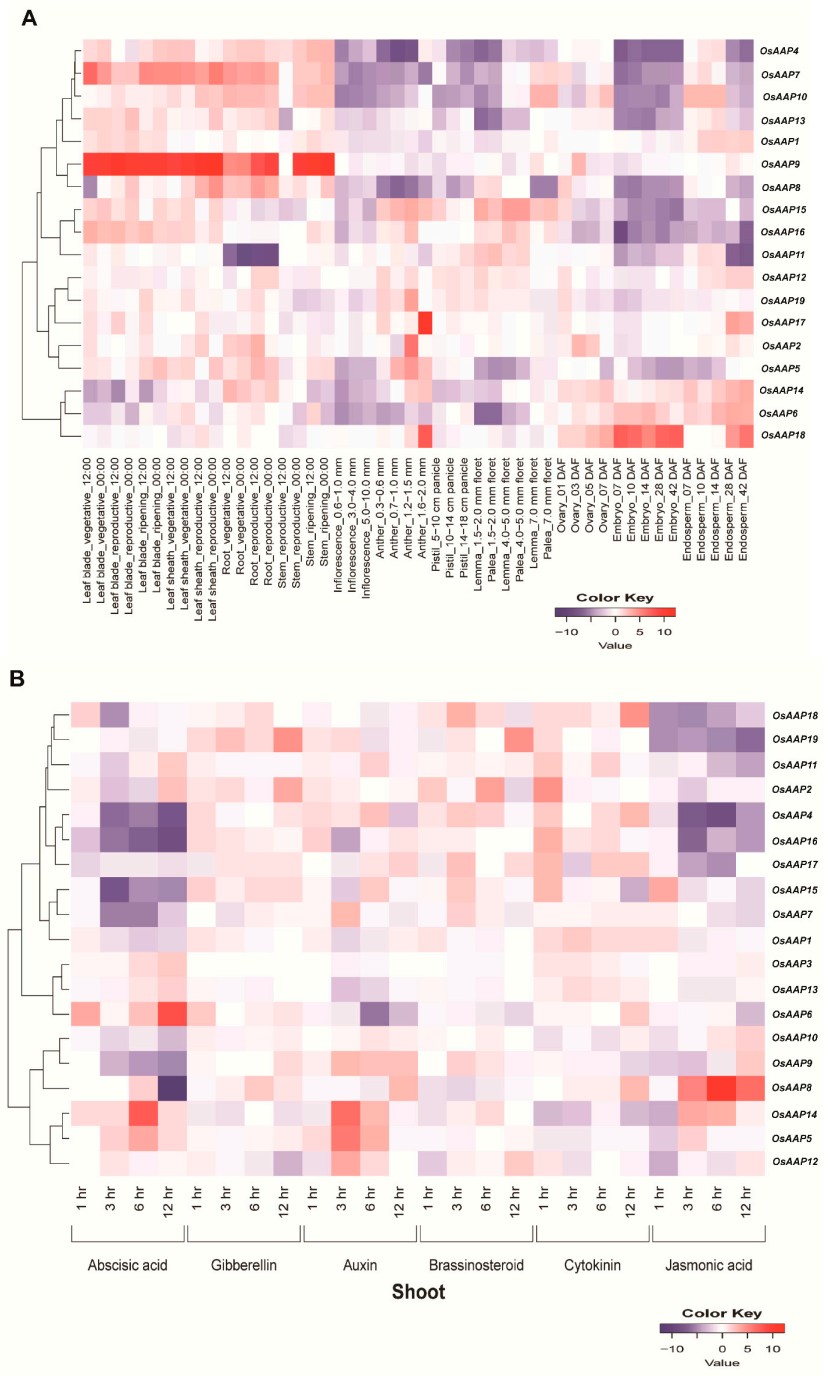

**Figure 9.** *Cont.*

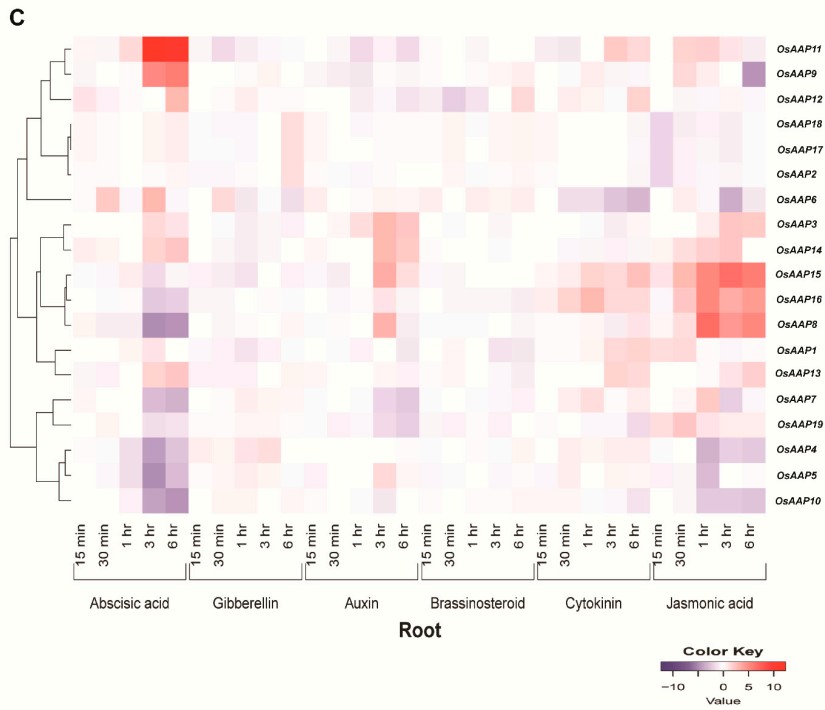

**Figure 9.** Heatmaps are utilized to visually represent the expression patterns of OsAAP genes across diverse developmental stages, tissues, and in response to phytohormone treatments. (**A**) The heatmap illustrates the expression profile across distinct tissues and developmental stages. (**B**) The heatmap displays the expression profile within shoots when subjected to phytohormone treatments. (**C**) The heatmap showcases the expression profile within roots following phytohormone treatments.

### 3.6. Networking between miRNAs and the AAP Gene Family

To potentially identify rice miRNAs, a collection of 308 mature miRNA sequences from indica rice was obtained from PmiREN (pmiren.com/download) (accessed on 22 August 2023). For the purpose of identifying potential miRNA targets within the *OsAAP* gene family members, the Plant Small RNA Target Analysis Server (psRNATarget) hosted at https://www.zhaolab.org/psRNATarget/analysis (accessed on 22 August 2023) was utilized. The miRNAs targeting the APP gene family members in rice, the psRNATarget tool revealed that only a subset of AAP gene family members were targeted by conserved miRNAs (Tables S8 and 2).

For instance, *OsAAP4*, *OsAAP8*, *OsAAP11*, and *OsAAP14* were found to be targeted by distinct families of miRNAs (OIn-miR397 and Oin-miRN1147). Among these, *OsAAP4* was predicted to be targeted by five members *Oin-miR397a*, *Oin-miR397b*, *Oin-miRN1147a*, *Oin-miRN1147b*, and *Oin-miRN1147*, leading to cleavage. Simultaneously, *OsAAP8* was projected to undergo cleavage inhibition by *Oin-miRN6152*, *Oin-miR482c*, *Oin-miRN2235b*, and *Oin-miRN2235c*. Similarly, *OsAAP11* was forecasted to be a target for cleavage by *Oin-miRN6120O*, *In-miRN6121*, *Oin-miRN6122O*, *Oin-miRN6127*, and *OIn-miRN6128*. Furthermore, *OIn-miR164a*, *OIn-miR164b*, *OIn-miR164c*, and *OIn-miR164d* were predicted to suppress the translation of *OsAAP22* (Table 2). These observations underscore the potential roles of these miRNAs in orchestrating the expression of *OsAAP* gene family members during various developmental phases in rice. Consequently, these findings provide valuable insights into the specific functions of these miRNAs in diverse biological processes in rice. Gaining a deeper understanding of the regulatory functions performed by these miRNAs can offer valuable insights into the molecular mechanisms governing rice development, stress responses, and various other essential biological processes. Thus, this knowledge could also hold practical implications for enhancing the rice productivity and resilience in the face of environmental challenges.

**Table 2.** List of miRNAs targeting various *OsAAPs* along with their respective modes of inhibition.

| Gene Name | Number of Targeting miRNA | miRNA | miRNA Sequence | Length (nt) | Inhibition |
|---|---|---|---|---|---|
| *OsAAP1* | 1 | OIn-miRN6163 | UUUUGAACGACUUGCACGAGA | 21 | Cleavage |
| *OsAAP2* | 2 | OIn-miR159a | CUUGGACUGAAGGGUGCUCCCU | 22 | Cleavage |
| | | OIn-miR159b | UUGGACUGAAGGGUGCUCCCU | 21 | Cleavage |
| *OsAAP4* | 5 | OIn-miR397a | UUGAGUGCAGCGUUGAUGAACC | 22 | Cleavage |
| | | OIn-miR397b | UUGAGUGCAGCGUUGAUGAAC | 21 | Cleavage |
| | | OIn-miRN1147a | CGUUCCCCAGCGGAGUCGCCA | 21 | Cleavage |
| | | OIn-miRN1147b | CGUUCCCCAGCGGAGUCGCCA | 21 | Cleavage |
| | | OIn-miRN1147c | CGUUCCCCAGCGGAGUCGCCA | 21 | Cleavage |
| *OsAAP5* | 2 | OIn-miR1850a | UGGAAAGUUGGGGAGAUUGGGG | 21 | Cleavage |
| | | OIn-miR166a | UCUCGGAUCAGGCUUCAUUCC | 21 | Cleavage |
| *OsAAP8* | 4 | OIn-miRN6152 | CCAGUGAAGAGUACUUUGGCU | 21 | Cleavage |
| | | OIn-miR482c | UUCCCGAUGCCUCCCAUGCCUA | 22 | Cleavage |
| | | OIn-miRN2235b | UUUUUUAAUAGAACCGACACCU | 22 | Cleavage |
| | | OIn-miRN2235c | UUUUUUAAUAGAACCGACACCU | 22 | Cleavage |
| *OsAAP11* | 5 | OIn-miRN6120 | UUGUUGUACUGUAUCAGCACCU | 22 | Cleavage |
| | | OIn-miRN6121 | UUGUUGUACUGUAUCAGCACCU | 22 | Cleavage |
| | | OIn-miRN6122 | UUGUUGUACUGUAUCAGCACCU | 22 | Cleavage |
| | | OIn-miRN6127 | UUGUUGUACUGUAUCAGCACCU | 22 | Cleavage |
| | | OIn-miRN6128 | UUGUUGUACUGUAUCAGCACCU | 22 | Cleavage |
| *OsAAP13* | 1 | OIn-miR1846b | UCCCACCGAGCAGCCGGAUCUC | 22 | Cleavage |
| *OsAAP14* | 6 | OIn-miR164a | UGGAGAAGCAGGGC-ACGUGCA | 21 | Cleavage |
| | | OIn-miR164b | UGGAGAAGCAGGGC-ACGUGCA | 21 | Cleavage |
| | | OIn-miR164c | UGGAGAAGCAGGGC-ACGUGCA | 21 | Cleavage |
| | | OIn-miR164d | UGGAGAAGCAGGGC-ACGUGCU | 21 | Cleavage |
| | | OIn-miR169l | AGCCAAGGAUGACUUGCCGGC | 21 | Cleavage |
| | | OIn-miRN2219a | AUCGGAGGCCAUGGUGCAGCC | 21 | Cleavage |
| *OsAAP15* | 1 | OIn-miRN6160 | UACCUCGGGCAACUGAAGACU | 21 | Cleavage |
| *OsAAP16* | 1 | OIn-miRN2274a | CACCAGGGAUUUCAUCGACUC | 21 | Cleavage |
| *OsAAP17* | 1 | OIn-miRN6160 | UACCUCGGGCAACUGAAGACU | 21 | Cleavage |
| *OsAAP20* | 1 | OIn-miRN6168 | UGCGAGGUUCACCAUGUUCUG | 21 | Translation |
| *OsAAP22* | 4 | OIn-miR164a | UGGAGAAGCAGGGCACGUGCA | 21 | Translation |
| | | OIn-miR164b | UGGAGAAGCAGGGCACGUGCA | 21 | Translation |
| | | OIn-miR164c | UGGAGAAGCAGGGCACGUGCA | 21 | Translation |
| | | OIn-miR164d | UGGAGAAGCAGGGCACGUGCU | 21 | Translation |
| *OsAAP23* | 3 | OIn-miRN2248a | CUUUUUCCUUGGGAAGGUGGU | 21 | Translation |
| | | OIn-miRN6133 | CGGUUCCUGUCCCAAGAUCGAG | 22 | Cleavage |
| | | OIn-miRN6138 | CGGUUCCUGUCCCAAGAUCGAG | 22 | Cleavage |
| *OsAAP24* | 1 | OIn-miR2118a | CUCCUGAUGCCUCCCAAGCCUA | 22 | Translation |
| *OsAAP25* | 1 | OIn-miR2118a | CUCCUGAUGCCUCCCAAGCCUA | 22 | Translation |
| *OsAAP27* | 4 | OIn-miR172a | AGAAUCUUGAUGAUGCUGCAU | 21 | Cleavage |
| | | OIn-miR172b | AGAAUCUUGAUGAUGCUGCAU | 21 | Cleavage |
| | | OIn-miR172c | AGAAUCUUGAUGAUGCUGCAU | 21 | Cleavage |
| | | OIn-miR172d | GGAAUCUUGAUGAUGCUGCAU | 21 | Cleavage |

### 3.7. Analysis of the Protein–Protein Interactions among the OsAAP Family Genes

Using the STRING database, a network was formulated to explore the protein–protein interactions involving OsAAPs and other rice proteins (Figure 10 and Table S9). Based on the predictive outcomes, we identified twenty-six OsAAPs that interact with eight different proteins specific to indica rice. Notably, OsAAP2 exhibits interactions with OsAAP9 and three other rice proteins, namely OsI_17768, OsI_02618, and OsI_32326. These rice proteins

encompass a V-type proton ATPase subunit, which is responsible for contributing to vacuolar ATPase (V-ATPase), a complex enzyme pivotal in acidifying intracellular organelles within eukaryotic cells [77,78]. Similarly, OsAAP4 engages in interactions with three distinct rice proteins: OsI_00709, OsI_32326, and OsI_02618. These proteins are characterized as an uncharacterized protein, nicotinate phosphoribosyl transferase, and a V-type proton ATPase subunit. Among these, nicotinate phosphoribosyltransferase is involved in catalyzing the conversion of nicotinate (Na) to nicotinate mononucleotide, marking the initial step in the Preiss–Handler pathway essential for NAD(+) biosynthesis [79,80]. The identified protein–protein interactions can contribute to a more profound comprehension of the intricate regulatory mechanisms involved in both biotic and abiotic stress responses, as well as the roles of *OsAAP* gene family members in growth and development processes. These findings present valuable insights that can propel further investigations into the functional characterization of *OsAAP* genes, shedding light on their roles and activities. Thus, this knowledge can ultimately aid in the development of high-yielding and stress-tolerant rice cultivars.

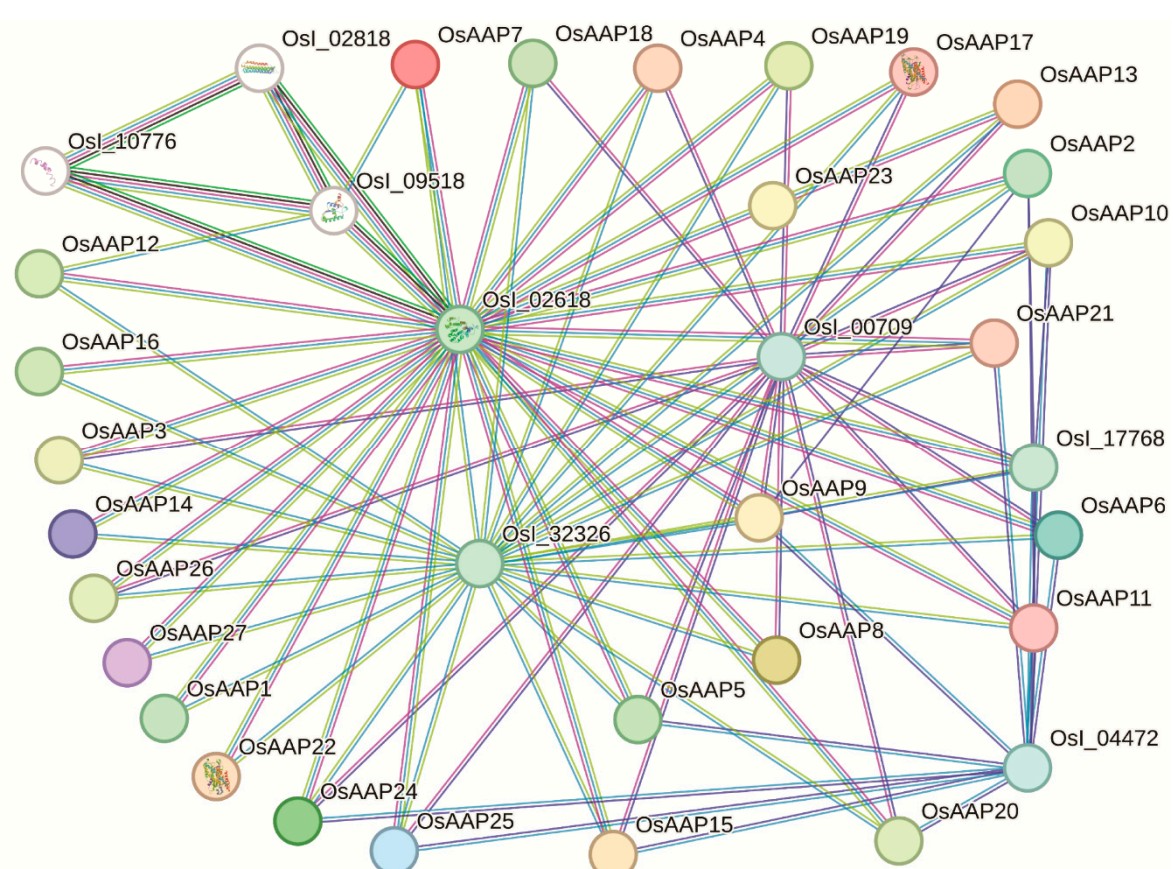

**Figure 10.** An exploration of protein–protein interactions among OsAAP proteins involves an analysis of their interconnections. The protein–protein interaction network is generated using STRINGV9.1. Here, individual proteins are represented as nodes, while interactions are depicted as edges. Furthermore, the edges are color-coded based on the type of evidence supporting the interaction.

### 3.8. Regression Analysis of OsAAP17 Gene with Culm Number in Rice

To identify the significant genetic variants in the *OsAAP17* gene associated with the culm number in rice, three regression models were used: linear, LASSO, and random forest. In the analysis, 389 genetic variants (SNPs and In/Dels) in 1340 rice genotypes from the 3 K database were used. The linear regression analysis revealed an adjusted $R^2$ value of 9.08% for the culm number contributed by the *OsAAP17* genetic variants in rice, with a

significant *p*-value ($2.812 \times 10^{-10}$) (Figure S7). Further, at the 5% level of significance, the linear regression model identified four variants (Chr06:21186420, Chr06:21187201, Chr06: 21188911.02, and Chr06: 21187038.01) that were significantly associated with the culm number. Similarly, LASSO and the random forest regression analysis revealed nine and five genetic variants associated with the rice culm number, respectively (Figure S8). The mean difference in the culm number for these 17 *OsAAP17* genetic variants revealed that all but one variant was significant for the mean difference in the culm number for the alternate alleles (Table 3).

**Table 3.** Mean difference of the SNPs of *OsAAP17* gene significantly associated with the culm number in rice using three different regression models.

| Sl.No | *OsAAP17* Gene SNP Ids | Mean Value of Culm Number of SNP1 (Nos.) | Mean Value of Culm Number of SNP2 (Nos.) | *p*-Value * |
|---|---|---|---|---|
| | | Linear Regression model | | |
| 1 | 21186420 | 17.62 | 15.56 | $2.22 \times 10^{-16}$ |
| 2 | 21187038 | 15.31 | 17.29 | $9.99 \times 10^{-16}$ |
| 3 | 21187201 | 17.31 | 16.40 | 0.004662 |
| 4 | 21188911 | 15.17 | 17.29 | 0 |
| | | LASSO | | |
| 1 | 21186274 | 16.46 | 19.91 | 0.003351 |
| 2 | 21186676 | 14.46 | 16.49 | 0.010262 |
| 3 | 21187004 | 17.32 | 14.69 | 0 |
| 4 | 21188056 | 17.32 | 14.98 | 0 |
| 5 | 21188057 | 16.43 | 18.5 | 0.001057 |
| 6 | 21188177 | 19.16 | 16.48 | 0.041175 |
| 7 | 21188591 | 16.49 | 21.5 | 0.030368 |
| 8 | 21189128 | 16.37 | 16.52 | 0.357885 [ns] |
| | | Random Forest | | |
| 1 | 21187038 | 17.98 | 15.55 | 0 |
| 2 | 21185768 | 16.49 | 18.68 | 0.035122 |
| 3 | 21188179 | 17.22 | 16.02 | $1.29 \times 10^{-6}$ |
| 4 | 21188056 | 17.32 | 14.98 | 0 |
| 5 | 21186226 | 15.29 | 17.25 | 0 |

* Mean difference statistically analyzed using Z test at 5% and 1% level of significance, ns—non-significant and Sl. No denotes the serial number.

### 3.9. Classification Analysis of Important Genetic Variants of OsAAP17 Gene

Using a classification analysis of random forest models, the identified seventeen variants in the *OsAAP17* gene were tested for their ability to differentiate between rice subpopulations (Table 4).

**Table 4.** Classification analysis of the rice ecotypes based on the genetic variants of *OsAAP17* gene.

| Ecotypes * | Admix | Aro | Aus | Indica | Japonica | Error Rate |
|---|---|---|---|---|---|---|
| admix | - | - | 11 | 15 | 9 | 1.0 |
| aro | - | - | 13 | 1 | 21 | 1.0 |
| aus | - | - | 119 | 1 | - | 0.008 |
| indica | 1 | - | 23 | 719 | 16 | 0.055 |
| japonica | - | - | 7 | 31 | 351 | 0.108 |

* Ecotypes are grouped in the rows and the values in the column indicate their classification based on the SNPs of *OsAAP17* gene.

The analysis revealed that the genotypes in the aus population have the smallest clarification error of 0.0083. With the exception of one, all genotypes in the aus subpopulation (119) were classified as aus sub-class in the analysis. Similarly, with a classification error of 0.055, 40 out of 761 indica genotype subpopulations were classified as other subpopulations

(aus and japonica). Further, with a clarification error rate of 0.108, 38 of 389 japonica genotypes were classified as aus and indica subpopulations. Furthermore, four variants in the gene's intronic region (*OsAAP17*: 21187038, 21189128, 21187004, 21187201) had a gini value decrease of more than 20. Furthermore, one variant in the coding sequence (*OsAAP17*:21191370) was discovered to have a high gini value of 18 (Figure S8).

*3.10. Diversity Analysis of OsAAP17 Gene in Rice Varieties and Aus Genotypes*

Five *OsAAP17* gene InDel markers were validated in 131 different indica rice varieties for the allelic data analysis (Figure S9). All five markers were found to be monomorphic in the rice varieties studied. The five *OsAAP17* InDels were then validated in 24 different aus genotypes and found to be polymorphic for three markers and monomorphic for two. For the combination of aus genotypes and rice varieties, the gene diversity was 0.0845 and the polymorphism information content was 0.0777 (Table 5).

**Table 5.** Gene diversity analysis of the five *OsAAP17* In/Del markers in different aus lines and rice varieties.

| Marker | Major Allele Frequency | Genotype No | Sample Size | No. of Obs. | Allele No | Availability | Gene Diversity | Hetero Zygosity | PIC |
|---|---|---|---|---|---|---|---|---|---|
| *OsAAP17_1* | 1.00 | 1.0 | 155 | 155 | 1.0 | 1.0 | 0.00 | 0 | 0.00 |
| *OsAAP17_2* | 1.00 | 1.0 | 155 | 155 | 1.0 | 1.0 | 0.00 | 0 | 0.00 |
| *OsAAP17_3* | 0.94 | 2.0 | 155 | 155 | 2.0 | 1.0 | 0.11 | 0 | 0.10 |
| *OsAAP17_4* | 0.87 | 2.0 | 155 | 155 | 2.0 | 1.0 | 0.21 | 0 | 0.19 |
| *OsAAP17_5* | 0.94 | 2.0 | 155 | 155 | 2.0 | 1.0 | 0.09 | 0 | 0.09 |
| Mean | 0.95 | 1.6 | 155 | 155 | 1.6 | 1.0 | 0.08 | 0 | 0.07 |

Furthermore, the gene diversity of the *OsAAP17_3*, *OsAAP17_4*, and *OsAAP17_5* markers was 0.1094, 0.2151, and 0.0979, respectively, and the polymorphism information content was 0.1034, 0.1920, and 0.0931, respectively. An unweighted pair group with an arithmetic averaging (UPGMA) analysis also revealed two major clusters, A and B. Cluster A is divided into two sub-clusters, A1 and A2, with one and five aus lines, respectively. Cluster B is further subdivided into B1 and B2, which are further subdivided into B1a, B1b, and B2a, B2b. B1a and B1b each have two and three aus genotypes. Furthermore, B2a has 13 aus lines, whereas B2b has the largest cluster of all rice varieties (Figure S10).

*3.11. Descriptive Statistics of Mapping Population between N22 and JR201 in Two Different Seasons*

The descriptive statistics were calculated for seven distinct traits in the mapping population across both seasons (Table 6).

**Table 6.** Descriptive statistics analysis of yield-related traits in JR201 and N22 mapping populations. $F_6$ recombinant inbred lines (RILs) were used with sample size of 48 (24 nos. of aus allele of *OsAAP17* and 24 nos. of indica allele of *OsAAP17*) and two parents (N22 and JR201).

| Traits | Mean | | Standard Error | | Median | | Mode | | Kurtosis | | Skewness | | Range | |
|---|---|---|---|---|---|---|---|---|---|---|---|---|---|---|
| Season | SI | SII | SI | SII | SI | SII | SI | SII | SI | SII | SI | SII | SI | SII |
| No. of Tillers (nos) | 10.2 | 8.71 | 0.48 | 0.44 | 10 | 8 | 8 | 6.33 | −0.04 | 1.22 | 0.27 | 1.22 | 9.8–10.5 | 7.59–9.83 |
| Panicle length (cm) | 20.18 | 21.49 | 0.63 | 0.45 | 21.35 | 21.4 | 21.7 | 20.96 | 0.56 | 2.09 | 0.08 | 0.52 | 20.55–21.06 | 21.35–21.64 |
| No. of Spikelet (nos) | 95.16 | 109.8 | 9.71 | 11.68 | 64 | 85.16 | 51 | 61.66 | 1.87 | 4.71 | 1.72 | 2.26 | 91.12–99.2 | 100.11–119.47 |
| Unfilled Grain (nos) | 23.52 | 27.45 | 3.56 | 4.42 | 15.5 | 18.16 | 9 | 13 | 5.04 | 12.05 | 2.26 | 3.32 | 22.2–24.83 | 43.76–49.57 |
| Filled Grain (nos) | 71.64 | 82.33 | 6.85 | 8.09 | 51.5 | 68.16 | 46 | 36 | 1.5 | 2.99 | 1.61 | 1.91 | 68.91–74.37 | 56.34–69.9 |
| Panicle weight (gm) | 1.78 | 1.84 | 0.18 | 0.16 | 1.33 | 1.54 | 0.96 | 1.003 | 6.93 | 3.95 | 2.27 | 2.03 | 1.71–1.84 | 1.78–1.90 |
| Single plant yield (gm) | 17.14 | 15.97 | 1.01 | 1.10 | 16.52 | 13.29 | 9.14 | 13.46 | 6.00 | 0.08 | 1.83 | 0.97 | 7.09–47.04 | 7.69–58.23 |

The average tiller number in the mapping population was found to be 10.18 in Season I (Kharif) and 8.71 in Season II (Rabi), with the difference being statistically significant ($p = 0.012$). Additionally, the mean panicle length was 20.81 cm in Season I and 21.49 cm in Season II. Season I had the most and least spikelets per panicle, with 40 and 296 respectively, and Season II had 46.33 to 400.33. Furthermore, the population's average filled and unfilled grains for season I were 71.64 and 23.52, respectively. Season II, on the other hand, had relatively higher unfilled grains, with a mean value of filled and unfilled grains of 82.33 and 27.45, respectively. Further, the mean difference in unfilled grains between seasons was found to be statistically significant ($p = 7.67 \times 10^{-6}$). Furthermore, the single plant yields for seasons I and II were 17.14 and 15.97 g, respectively.

### 3.12. Allelic Effect of the Alleles of the OsAAP17 Gene for Yield-Related Traits

The recombinant inbred lines (RILs) of the F6 mapping populations were created by crossing two parents, N22 and JR201. The two parents were initially genotyped using five *OsAAP17* gene In/Del markers. Only one marker (*OsAAP17-5*) was found to be polymorphic between the parents, while the remaining four markers were found to be monomorphic between the two parents. The *OsAAP17-5* marker was discovered to have two allelic variants of amplicon size, 480 bp (JR201) and 500 bp (N22). In two different seasons, 480 bp (24 Nos.) and 500 bp (24 Nos.) mapping populations were evaluated for yield-related traits. Initially, the mean values of various yield-related traits were compared to see if there was a statistical difference between Season I and Season II (Table 7).

**Table 7.** Mean analysis of different yield-related traits between *OsAAP17* alleles in the mapping populations between JR201 and N22 in Season I.

| Traits | Mean Values of *OsAAP17* Alleles of Season I | | *p*-Value |
|---|---|---|---|
| | JR201 Allele (480 bp) | N22 Allele (500 bp) | |
| Tiller number (Nos.) | 9.8 ± 3.08 | 10.5 ± 3.71 | 0.262629 |
| Panicle length (cm) | 20.55 ± 4.43 | 21.06 ± 4.46 | 0.346168 |
| Number of spikelet (Nos.) | 91.12 ± 68.31 | 99.2 ± 67.54 | 0.340095 |
| Unfilled grain (Nos.) | 22.2 ± 23.28 | 24.83 ± 26.52 | 0.357805 |
| Filled grain (Nos.) | 68.91 ± 50.79 | 74.37 ± 44.89 | 0.346624 |
| Single panicle weight (g) | 1.84 ± 1.48 | 1.71 ± 1.008 | 0.358779 |
| Single plant yield (g) | 16.58 ± 5.11 | 17.7 ± 8.58 | 0.291189 |

This analysis revealed that the mean value of two traits, tiller number and unfilled grains, differed significantly between seasons (Figure 11A). Furthermore, the effects of two *OsAAP17* alleles for the primer (*OsAAP17*-5) on the various yield-related traits were investigated (Figure 11B,C, Tables 8 and 9).

**Table 8.** Mean analysis of different yield-related traits between *OsAAP17* alleles in the mapping populations between JR201 and N22 in Season II.

| Traits | Mean Values of *OsAAP17* Alleles of Season II | | *p*-Value |
|---|---|---|---|
| | JR201 Allele (480 bp) | N22 Allele (500 bp) | |
| Tiller number (TN) | 7.59 ± 2.4 | 9.8 ± 3.2 | 0.005095 ** |
| Panicle length (PL in cm) | 21.64 ± 3.31 | 21.35 ± 2.9 | 0.464042 |
| Number of spikelet (NOS) | 119.47 ± 94.19 | 100.11 ± 65.69 | 0.254926 |
| Number of unfilled grain (NUG) | 49.57 ± 32.01 | 43.76 ± 22.7 | 0.288966 |
| Number of Filled grain (NFG) | 69.9 ± 62.20 | 56.34 ± 42.96 | 0.238248 |
| Single panicle weight (SPW in g) | 1.9 ± 1.18 | 1.78 ± 1.1 | 0.404013 |
| Single plant yield (SPY in g) | 15.97 ± 7.73 | 17.68 ± 10.39 | 0.279943 |

TN: Tiller number; PL: Panicle length (cm); NOS: Number of spikelet; NUG: Unfilled grain; NFG: Filled grain; SPW: Single panicle weight (g); SPY—single-plant yield. Double asterisk indicates 1% level of significance.

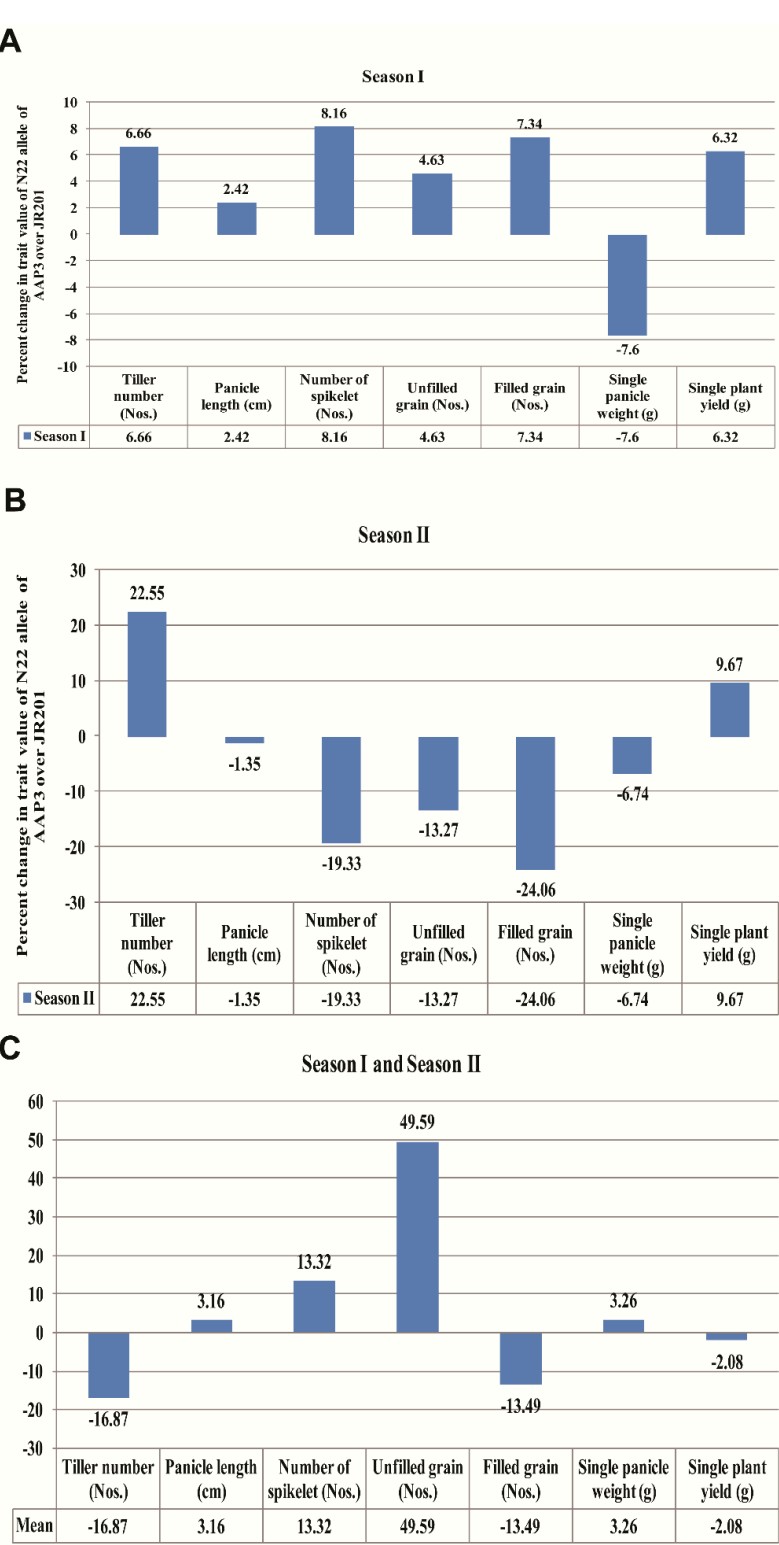

**Figure 11.** The percentage difference in mean yield-related traits between the N22 allele of *OsAAP17* and JR201 was assessed within the mapping population. (**A**) The percentage difference in yield-related traits of the N22 allele of *OsAAP17* compared to JR201 is assessed within the mapping population during Season I. (**B**) The percentage difference in yield-related traits between the N22 allele of *OsAAP17* and JR201 is analyzed within the mapping population during Season II. (**C**) The percentage difference in mean yield-related traits between Season II compared to Season I within the mapping population for both seasons.

**Table 9.** Mean analysis of different yield-related traits in two different seasons in the mapping populations between JR201 and N22.

| Traits | Mean Values in Season I and Season II | | *p*-Value |
| --- | --- | --- | --- |
| | Season I | Season II | |
| Tiller number (TN) | $10.18 \pm 3.39$ | $8.71 \pm 3.07$ | 0.012941 * |
| Panicle length (PL in cm) | $20.81 \pm 4.41$ | $21.49 \pm 3.12$ | 0.189431 |
| Number of spikelet (NOS) | $95.16 \pm 67.32$ | $109.79 \pm 80.93$ | 0.16791 |
| Number of unfilled grain (NUG) | $23.52 \pm 24.72$ | $46.66 \pm 30.63$ | $7.67 \times 10^{-6}$ ** |
| Number of Filled grain (NFG) | $71.64 \pm 47.50$ | $63.12 \pm 56.08$ | 0.204173 |
| Single panicle weight (SPW in g) | $1.78 \pm 1.25$ | $1.84 \pm 1.13$ | 0.403848 |
| Single plant yield (SPY in g) | $17.14 \pm 7.01$ | $16.82 \pm 7.64$ | 0.062107 |

TN: Tiller number; PL: Panicle length (cm); NOS: Number of spikelet; NUG: Unfilled grain; NFG: Filled grain; SPW: Single panicle weight (g); SPY—single-plant yield. Double asterisk indicates 1% level of significance and single asterisk indicates significance at 5% level.

In Season I, the mean difference for the different traits for both *OsAAP17* alleles in the mapping population was found to be non-significant. In season I, the aus allele of the *OsAAP17* gene had slightly higher trait values for the tiller number, number of spikelets per panicle, number of filled grains per panicle, panicle length, and single plant yield. The mean difference in productive tillers per plant was found to be significant (*p* = 0.005) in season II. The average productive tiller number for the JR201 (480 bp) and N22 (500 bp) allele lines was 7.59 and 9.8, respectively. Furthermore, the allelic difference between the JR201 (480 bp) and N22 (500 bp) alleles of the *OsAAP17*-5 marker for the number of productive tillers was 2.21, and lines with the N22 allele contributed 22.55% more productive tillers than lines with the JR201 allele. Further, the percent difference in the number of spikelets was higher for the lines carrying the JR201 allele of *OsAAP17*, but the mean difference was not statistically significant. Similarly, the JR201 allele had a high percentage of unfilled grains, but it was not significantly different from the *OsAAP17* N22 allele. Furthermore, in season II, the N22 allele of *OsAAP17* yielded 9.67% more than the JR 201.

## 4. Discussion

Amino acid transporters serve a crucial role in the transmembrane transportation of amino acids, which play direct or indirect roles in nitrogen metabolism processes vital for plant growth and development [1,2,26,81]. These processes encompass the cell's assimilation and allocation of amino acids, as well as the movement of amino acids across short and long distances, along with the absorption and utilization of amino acids by sink organs [3,6]. Recently, the AAP gene family has gained considerable attention among plant scientists and has emerged as a pivotal area of research focus. These transporters play roles in processes such as seed filling, transferring amino acids between the xylem and phloem, loading amino acids into the phloem, and absorbing amino acids from the soil into plants [7]. In addition, recent investigations have indicated that elevating the loading of amino acids into the phloem and embryos could potentially enhance the biomass and seed yield [6,24]. Although its importance is recognized, a comprehensive grasp of this family's dynamics in indica rice remains lacking. In this study, a comprehensive identification revealed a total of 27 AAP family genes in the genome of indica rice. Phylogenetic analysis unveiled that the 69 AAP genes from both the model species and other plant species could be classified into 16 distinct subfamilies. The analysis of cis-elements within *OsAAP* genes revealed that the promoters of *OsAAP* encompass both phytohormone and plant growth and development, as well as stress-related cis-elements. Additionally, concerning transcriptome profiling, a significant portion of the genes showed responsiveness to various hormones, and their activation spanned a range of tissues and developmental stages in rice. Furthermore, the gene diversity for the *OsAAP17* gene was comparatively higher in the aus genotypes when

compared to the rice varieties that have already been released. The identification of an *OsAAP17* allele resulted in an increase in both the productive tiller count and overall yield. The detailed implications and significance of these findings are thoroughly discussed below.

### 4.1. The OsAAP Gene Family in Rice: Identification, Characterization, and Evolutionary Analysis

Amino acid transporters play a critical role in seed filling, mediating the transfer of amino acids between the xylem and phloem, facilitating amino acid loading into the phloem and enabling the uptake of amino acids from the soil into plants [1,2,26,81]. In this study, a comprehensive analysis led to the identification of a total of 27 AAP family genes in the indica rice genome. The phylogenetic analysis unveiled that the 69 AAP genes from both the model species and other plant species could be classified into 16 distinct subfamilies (Figure 1). Among these, Group XV comprised eight members, while Groups I, II, III, IV, V, VI, VII, VIII, IX, X, XI, XII, XIII, XIV, and XVI contained two, two, one, zero, zero, one, one, one, zero, one, two, zero, one, four, and three members, respectively (Figure S2). The phylogenetic analysis suggests a notable level of evolutionary expansion in the *OsAAP* family in rice. Further, predictions regarding subcellular localization indicated that 22 OsAAP proteins were situated in the plasma membrane, while the rest were distributed across the chloroplast outer membrane (OsAAP16), the endomembrane system (OsAAP11 and OsAAP19), and organelle membranes (OsAAP9 and OsAAP23). A correlation between the molecular weight and pI of OsAAP was examined to establish the distribution pattern of OsAAP proteins (Figure S1). These findings unveiled a similarity in molecular weight and the isoelectric point across nearly all OsAAP proteins. The chromosomal mapping of the identified *OsAAP* family genes revealed that the distribution of the 27 *OsAAP* genes across the 12 rice chromosomes was uneven (Figure 2). Chromosomes 1 and 4 exhibited the highest count of *OsAAP* genes, each containing five genes, while chromosome 9 contained the fewest with one gene. Our findings indicate that gene duplication influences the chromosomal positioning of *OsAAP* genes, and the expansion of the gene family is contingent upon sequence duplication, whether it occurs via whole genome duplication (WGD) or segmental events. Broadly, gene duplication assumes a significant role in driving the expansion and evolution of gene families [82]. Tandem repeats give rise to gene clusters or hotspots, while fragment repeats yield homologous genes, both contributing to this process [83]. In this study, a total of 10 duplicated gene pairs were identified in the *OsAAP* gene family (Figures 3 and S3). The Ka/Ks scores for these genes were lower than 1, indicating a substantial purifying selection with slight changes subsequent to gene duplication. Nevertheless, a solitary gene pair (*OsAAP10-OsAAP11*) displayed a Ka/Ks value exceeding one, suggesting a positive selection effect on this particular gene pair. These insights suggest that the growth of the AAP gene family in rice has predominantly stemmed from both segmental and whole-genome duplications (Table S3). The genome conservation was visualized via the utilization of the Circoletto Tool (tools.bat.infspire.org/circoletto/) (accessed on 22 August 2023) in the process of conducting a comparative synteny analysis. The gene sequence of *OsAAP1* exhibited synteny with the gene sequence *ONIVA02G30080* from *O. nivara*. Similarly, the *HORVU.MOREX.r3.2HG0174620* gene from potatoes demonstrated synteny with *OsAAP7* in indica rice. Notably, the gene sequence of *OsAAP* exhibited synteny across these plant species, as depicted in Figure 4. These findings collectively showcase that the *OsAAP* family genes have experienced a conserved evolutionary process. The analysis of exon and intron structures was conducted on the genes of the *OsAAP* family to comprehend the diversity in their structural composition and how it contributes to the evolution of these family genes. Based on the gene structure analysis, it was evident that the quantity of introns differed across various subfamilies (Figure 5). In the spectrum of AAP subfamily genes, the count of introns spanned from 0 to 7; however, the majority exhibited between 2 to 3 introns. Notably, *OsAAP23* featured the highest count of seven introns, whereas *OsAAP7*, *OsAAP11*, *OsAAP12*, and *OsAAP26* displayed an absence of introns (Figures 5 and S4). Numerous studies have demonstrated that the loss of introns within plant gene families is a phenomenon that has occurred during

the course of plant evolutionary processes [84–89]. Intriguingly, it was observed that none of the AAP genes possessed 3'- and 5'-untranslated regions (UTRs). Further, for an in-depth exploration of the attributes inherent to OsAAP proteins, we successfully pinpointed 15 conserved motifs within these proteins (Figure 6 and Table S4). In this investigation, it was found that Motifs 1, 2, 3, 4, and 5 displayed a high level of conservation across most of the OsAAP proteins. In addition, through the alignment of amino acid sequences, it became apparent that OsAAP proteins incorporate these conserved motifs (Figures 7 and S5A). Furthermore, we conducted predictions for the 3D structure and transmembrane helices of OsAAP proteins (Figure S5B,C). As a result, while certain motifs exhibit considerable conservation across the OsAAP family, distinct subfamilies showed distinctive motifs that might contribute to specialized functions.

### 4.2. AAP Gene Family Encompasses Multiple CAREs and Implicated in Plant Growth and Developmental Processes in Rice

Cis-elements within the promoter region hold a pivotal role in governing gene expression regulation [90–92]. In this study, the CAREs analysis revealed numerous cis-regulatory elements encompassing aspects such as phytohormone responses, light sensitivity, circadian rhythm modulation, seed-specific regulation, as well as defense and stress responsiveness (Figure 8A,B). Regarding hormonal responses, the analysis revealed the presence of various responsive elements such as the ABRE, AuxRE, GARE, SARE, and MeJARE (Table S5). In the promoter regions of most genes, we identified the presence of MeJARE and ABRE elements. Notably, light-responsive, MeJARE, and ABRE elements were prevalent across nearly all *OsAAP* genes. However, the SARE, AuxRE, and GARE-motif elements were only detected in a limited number of promoter regions among the *OsAAP* genes. In stress response, there are elements related to anaerobic induction, drought induction, low temperature response and salt stress, defense and stress responses. The study examined the expression patterns of *OsAAP* family genes across various tissues, developmental stages, and in response to hormone treatments. Remarkably, 27 *OsAAP* family genes displayed varying expression levels across distinct tissues, developmental stages, and in response to the application of hormones (Figure 9A–C). For example, the expression of *AAP7* and *AAP9* was heightened in various stages: vegetative, reproductive, and ripening leaf blades, as well as in vegetative and reproductive leaf sheaths and vegetative and reproductive roots. Conversely, the induction of an *AAPX18* expression was observed in the anther ovary, embryo, and endosperm. Further, an *AAPX17* expression is prominently stimulated in the anther and endosperm. In contrast, *AAP15* exhibits significant up-regulation in the anther, lemma, and palea. Furthermore, we conducted an analysis of the expression profile of *OsAAP* in both shoots and roots, subjecting them to various plant hormone treatments including gibberellin (GA), cytokinin, auxin, abscisic acid (ABA), jasmonic acid (JA), and brassinosteroid (BRS) (Figure 9B,C). In the shoot, the expression of *OsAAP5* and *OsAAP14* was elevated following both ABA and auxin treatments after 3 h and 6 h. Meanwhile, *OsAAP8* displayed an increase in the expression after 3 h, 6 h, and 12 h of auxin treatment in the shoots. Additionally, the expression of *OsAAP2* and *OsAAP19* increased under the GA treatment after 1 h, 3 h, 6 h, and 12 h in the shoots (Figure 9B). Further, the expression of *OsAAP8*, *OsAAP15*, and *OsAAP16* exhibited an increase in the root under the JA treatment at 30 min, 1 h, 3 h, and 6 h. Similarly, for *OsAAP15* and *OsAAP16*, the expression levels were elevated in response to the cytokinin treatment at 30 min, 1 h, 3 h, and 6 h in the root. After 3 h and 6 h of auxin treatment in the root, the expression levels of *OsAAP3*, *OsAAP14*, and *OsAAP15* were induced. Following 3 h and 6 h of ABA treatments in the root, the transcript levels of *OsAAP9* and *OsAAP11* were observed to increase (Figure 9C). These findings indicate that the various members of the *OsAAP* gene family could be involved in distinct developmental processes, stress response, and might respond to phytohormones in rice. The overexpression of *PtAAP1* has been shown to enhance plant nitrogen use efficiency (NutE) by altering the transport of amino acids from the source to the sink in pea plants [24]. In rice, another member of the NPF family, *OsNPF7.3*, which transports

dipeptides and tripeptides like Gly-His-Gly and Gly-His, influences the rice tiller number and NutE [93]. A recent study has revealed that it plays a crucial role in influencing growth and the grain yield in rice by controlling the absorption and distribution of neutral amino acids [94]. It is well established that amino acid transport is extensively modulated via environmental cues, including factors like drought, low temperature, light, and high salt [95]. Notably, the expression of *AtAAP4* and *AtAAP6* has been documented to be downregulated under salt stress in Arabidopsis [96], and similar observations were made for *McAAT2* in *M. crystallinum* [97]. *AtAAP3*, which is primarily expressed in root vascular tissue, could potentially be linked to amino acid uptake from the phloem [16], while *AtAAP6* could potentially be accountable for the uptake of amino acids from the xylem [15]. Considering their phylogenetic relationship and the roles of these *AtAAPs*, we hypothesize that *OsAAP6*, *OsAAP21*, and *OsAAP26* could be engaged in amino acid uptake and long-distance transport. Previous research has shown that *AtAAP8* participated in amino acid uptake into the endosperm and supplies amino acids to the developing embryo during early embryogenesis [19]. It is plausible to speculate that these members of the AAP family also hold vital roles in nutrient transport during seed development in rice. Multiple lines of evidence suggest the presence of interactions between developmental processes and stress responses, where specific genes may be co-regulated via both environmental factors and developmental signals [98]. Notably, there are reports of a network of rice genes that are intertwined with stress responses and seed development [99]. The Gene Ontology (GO) enrichment analysis also suggested the vital involvement of *OsAAP* family genes in diverse cellular processes in rice (Figure S6, Tables S6 and S7). In addition, we also identified potential rice miRNAs miRNA targets in the *OsAAP* gene family members (Tables S8 and S9). For instance, *OsAAP4*, *OsAAP8*, *OsAAP11*, and *OsAAP14* were found to be targeted by distinct families of miRNAs (OIn-miR397 and OIn-miRN1147). Among these, *OsAAP4* was predicted to be targeted by five members, *OIn-miR397a*, *OIn-miR397b*, *OIn-miRN1147a*, *OIn-miRN1147b*, and *OIn-miRN1147*, leading to cleavage. Simultaneously, *OsAAP8* was projected to undergo cleavage inhibition by *OIn-miRN6152*, *OIn-miR482c*, *OIn-miRN2235b*, and *OIn-miRN2235c*. Similarly, *OsAAP11* was forecasted to be a target for cleavage by *OIn-miRN6120O*, *In-miRN6121*, *OIn-miRN6122O*, *OIn-miRN6127*, and *OIn-miRN6128*. Furthermore, *OIn-miR164a*, *OIn-miR164b*, *OIn-miR164c*, and *OIn-miR164d* were predicted to suppress the translation of *OsAAP22* (Table S9). These discoveries offer valuable insights into the specific functions of these miRNAs in diverse biological processes in rice. Deeper comprehension of the regulatory functions of these miRNAs can enrich our understanding of the molecular underpinnings governing rice development, stress responses, and other pivotal biological mechanisms. Moreover, the protein–protein interaction outcomes revealed that twenty-six *OsAAPs* interact with eight different proteins specific to indica rice (Figure 10 and Table S10). Notably, OsAAP2 exhibits interactions with OsAAP9 and three other rice proteins, namely OsI_17768, OsI_02618, and OsI_32326. These rice proteins encompass a V-type proton ATPase subunit, which is responsible for contributing to vacuolar ATPase (V-ATPase), a complex enzyme pivotal in acidifying intracellular organelles within eukaryotic cells [77,78]. Similarly, OsAAP4 engages in interactions with three distinct rice proteins: OsI_00709, OsI_32326, and OsI_02618. These findings offer valuable insights that can propel further investigations into the functional characterization of *OsAAP* genes, shedding light on their roles and activities. Collectively, these results indicate that the members of the *OsAAP* gene family could potentially engage in diverse developmental processes and respond to phytohormones in rice. Consequently, the outcomes of this study hold significant promise in unraveling the biological functions of *OsAAP* gene family members in the future.

### 4.3. Aus Rice Has a Unique Genotypic Constitution for the OsAAP17 Gene in Rice

The implementation of a classification analysis utilizing a random forest approach yielded the lowest classification error rate for the aus ecotypes of rice (0.8%), in contrast to the India (5.5%) and japonica (10.8%) ecotypes. This outcome underscores the distinctive

genetic makeup of the *OsAAP17* gene within the aus ecotypes of rice. Prior research has also employed a random forest classifier to categorize rice genotypes into various subpopulations, including India and Japonica groups, via the utilization of a set of 268 predictive SNPs. These SNPs are likely to have been subjected to selection pressures and are distributed across multiple chromosomes within the rice genome [75]. Among the total of 389 SNPs present in *OsAAP17*, seventeen were found to potentially be under the influence of selection pressures, which could impact the classification as well as the trait responses across distinct rice ecotypes. Additionally, a previous investigation identified differences in the promoter regions between indica and japonica rice ecotypes for both the *OsAAP17* [27] and AAP5 genes [100]. Interestingly, the current study highlighted that a majority of the predictive SNPs are situated within the first intron region of *OsAAP17*. This observation suggests that functional divergence of *OsAAP17* introns has occurred across various rice ecotypes, including indica, aus, and japonica. Thus, understanding the implications of such intron divergence on the gene expression and the regulation of traits is imperative.

### 4.4. Aus Allele of OsAAP17 Enhances the Tiller Number and Yield in Rice

Aus-type rice, a distinct variety within the realm of rice, has garnered recognition as a valuable genetic reservoir for conferring tolerance to abiotic stresses. Despite their close relationship with indica rice species, aus varieties are categorized as a separate and discernible group within the broader rice ecotype phylogenetic classification [101]. These landraces have evolved in regions with nutrient-deficient soils and continue to be cultivated under challenging environmental stress conditions in India and Bangladesh [102]. Nagina22 (N22), an aus-type rice variety utilized as a parental contributor in this study's development of mapping populations, has previously gained recognition as one of the most resilient genotypes in terms of heat and drought tolerance [103,104]. In this study, the presence of the N22 allele of *OsAAP17* was found to enhance both the tiller number and yield during the dry season, resulting in a respective increase of 22.7% and 9.6%. Notably, previous research has demonstrated that the down-regulation of *OsAAP3* using genome editing techniques such as CRISPR and RNA interference led to a substantial increase in the number of tillers by stimulating bud outgrowth. This manipulation also resulted in enhanced yield and greater grain filling percentage, exhibiting an improvement of around 35–40% in japonica rice varieties [27]. This observation implies that the variation in trait effects might stem from the distinct expression patterns of *OsAAP3* in the two parental varieties used in this study, namely N22 and JR201. In support of this hypothesis, Lu et al. [27] presented evidence that the expression of AAP3 in indica genotypes exhibited a 3–4 fold reduction compared to japonica genotypes, a pattern that correlated with a higher tiller count in Indian genotypes. Furthermore, Lu et al. [27] also highlighted the existence of major haplotypes specific to rice ecotypes (indica, aus, and japonica), suggesting that the expression dynamics of AAP3 might indeed be ecotype-dependent in rice. Therefore, the adoption of the aus-specific in/del of *OsAAP17* within this study indirectly signifies the choice of the ausN22 *OsAAP17* haplotype for enhancing traits. This approach aligns with prior discoveries of superior aus-specific haplotypes for low phosphorus tolerance, low nitrogen tolerance, and grain iron content [105,106]. Consequently, the specific *OsAAP17* gene in/dels identified and confirmed in this study hold the potential to be integrated into rice haplotype breeding initiatives. Additionally, it is worth noting that the downregulation of *OsAAP3* has been shown to enhance nitrogen use efficiency (NUE) by 10% in japonica rice [27]. Thus, understanding the role of the aus *OsAAP17* allele in regulating NUE could offer valuable supplementary insights.

### 4.5. Environmental Effect of the Aus OsAAP17 Allele in Rice

A significant outcome of this study was the discernible allelic impact of the *OsAAP17* gene on both the yield and the number of productive tillers. Notably, this effect exhibited greater prominence and statistical significance during the rabi season (January to May) as compared to the Kharif season (August to December) in rice cultivation. The relative

percentage alteration in the count of productive tillers between the various *OsAAP17* alleles was recorded as 6.6% and 22.5% during the Kharif and rabi seasons, respectively. Previous research in the field of rice has noted that the interaction between temperature and light conditions can have an impact on the quantity of productive tillers [107,108]. While the role of *OsAAP3* gene expression variation in regulating the tiller number has been established for japonica rice [27], further exploration is needed to comprehensively grasp the effects of environmental factors on the expression of the *OsAAP17* gene.

### 4.6. Scope of Aus OsAAP17 Allele for Enhancing the Genetic Gain in Indica Rice Varieties

The screening of approximately 131 indica rice varieties that have been commercially cultivated in India unveiled a consistent presence of indica haplotypes within the studied genotypes. Interestingly, the superior aus in/del or haplotypes of the *OsAAP17* gene were absent in all examined varieties. This points toward the unconscious selection and establishment of indica-specific *OsAAP17* haplotypes via the breeding process conducted by rice breeders in India. Notably, in a parallel discovery (currently in preparation for publication), a significant gene governing both the yield and nitrogen use efficiency, *GRF4*, similarly displayed the prevalence of indica haplotypes among the released rice varieties. This pattern underscores the notion that incorporating the aus alleles of the *OsAAP17* gene could potentially enhance the tiller number and nitrogen use efficiency within indica rice varieties. A comparable observation was made with specific SNPs found exclusively in the *NCED2* gene of rice, which were fixed solely within upland rice varieties [109], and favorable allele enhanced aerobic adaptation [110]. Further, Xie et al. [111] findings highlight how foundational cultivars like IR8 exert a significant influence on the nucleotide diversity within specific genes or genomic regions among indica rice varieties. This prompts the consideration that founder cultivars might have contributed to the diversity observed in the *OsAAP17* gene. Assessing the introduction of the aus allele becomes essential for evaluating its potential contribution to genetic yield enhancements in rice varieties. Moreover, the inclusion of aus-type rice in breeding programs is highly recommended due to its favorable traits of stress tolerance. For instance, N22, an aus cultivar, has been a source of various heat and drought-tolerant quantitative trait loci (QTLs), which have been identified and incorporated into rice improvement initiatives [104]. These present findings establish a connection between the aus allele of the *OsAAP17* gene and improved yield outcomes.

### 4.7. Intron Diversity of OsAAP17 Gene and Functional Relationship

The aus allele of the *OsAAP17* gene shows associations with multiple SNPs related to the culm number. These SNP associations are located within the first intron of the gene. This suggests that intronic polymorphisms in *OsAAP17* could potentially impact the gene expression and play a role in regulating phenotypic traits. This observation is supported by previous findings indicating that introns situated near promoters can influence the expression of genes like potato Sus3 and Arabidopsis histone 3 (AtH3). Additionally, several instances of introns influencing the spatial expression in different genes have been observed, including examples such as Petunia ADF1, Arabidopsis PRF2, tomato sucrose transporter LeSUT1, and rice a- and b-tubulin genes. Thus, the variations within the intron of the *OsAAP17* gene might also contribute to the intron-dependent variability in the expression seen in rice.

### 4.8. Global Polymorphic In/Dels for Ecotype-Specific Haplotype-Introgression Breeding in Rice

The validation of the intron in/del of the *OsAAP17* gene for association with the tiller number and yield indicates that the approach of designing primers based on global diversity could be a valuable strategy for the PCR-based ecotype-specific haplotype-introgression breeding in rice. This method also proved effective in identifying specific insertions and deletions (in/dels) within the 3′ untranslated region (3′UTR) of the TPP7 gene in rice, which is associated with the response to anaerobic germination [72]. While rice improvement ini-

tiatives centered around haplotypes are gaining traction in popularity [112,113], employing the sequence data from the 3 k rice project with its substantial ~32 million variants poses certain challenges. The substantial quantity of haplotypes associated with each gene, along with the presence of numerous SNPs within the superior haplotypes, presents significant complexities for their practical utilization in introgression breeding methods. Consequently, the strategy proposed here, which entails the deliberate design of insertions and deletions (in/dels), while taking global variations into account, emerges as a promising approach for targeted introgression breeding programs focused on particular ecotypes.

**5. Conclusions**

In this study, a comprehensive total of 27 AAP genes were identified in the genome of indica rice. The phylogenetic analysis unveiled that the 69 AAP genes from both the model species and other plant species could be classified into 16 distinct subfamilies. Further, the examination of chromosomal mapping indicated an uneven distribution of the 27 *OsAAP* genes across the 12 rice chromosomes. The analysis of cis-elements within *OsAAP* genes revealed that the promoters of *OsAAP* encompass phytohormone and plant growth and development, as well as stress-related cis-elements. Additionally, transcriptome profiling revealed that most of the *OsAAP* genes exhibited responsiveness to different hormones, with their activation occurring across a spectrum of tissues and developmental stages in rice. The study identified miRNAs with a specific affinity for OsAAP genes. Out of the 27 OsAAP genes investigated, seventeen were discovered to be targeted by a total of forty-three miRNAs. The presence of the aus allele of the *OsAAP17* gene affects the tiller count in the N22 and JR 201 populations. The genetic variability present in the intron region of *OsAAP17* is likely to govern the gene expression, thereby indirectly impacting the tiller quantity and bud outgrowth in rice plants. Further research focused on examining how intron polymorphism influences the expression of the aus allele of *OsAAP17* would offer more profound insights into the transcriptional regulation of the *OsAAP17* gene. Moreover, to acquire a comprehensive understanding of the underlying molecular mechanisms, it is crucial to conduct an in-depth investigation into the environmental factors that regulate the expression of the *OsAAP17* gene. The discoveries presented here lay a strong foundation for further exploration into the roles of *OsAAP* family genes across a range of developmental processes. Therefore, the identified allelic variations in the utilization of *OsAAP17* has the potential to enhance rice crop production via molecular breeding in the changing climate scenario.

**Supplementary Materials:** The following are available online at https://www.mdpi.com/article/10.3390/agronomy13102629/s1, Figure S1: Plots comparing the molecular weights (kDa) and iso-electric points of OsAAP genes; Figure S2: Distribution of *OsAAPs* within a distinct cluster of the phylogenetic tree; Figure S3: An analysis of the phylogeny of *OsAAP* genes was conducted. A phylogenetic tree was generated using MEGAX via the neighbor-joining (NJ) approach, supplemented with 1000 bootstrap replications. Duplicated genes are denoted by a black asterisk; Figure S4: The structural organization of exons and introns within the OsAAP gene family; Figure S5: Alignment and three-dimensional structure depiction of the OsAAP protein sequences. A. The conserved protein tyrosine kinase domain is highlighted in red within a box. Chemically similar amino acids are indicated with colors and shading. Dashes signify gaps introduced to optimize alignment in the homologous region. B. The anticipated 3D structures of OsAAP proteins. C. The anticipated transmembrane helices within the OsAAP proteins were determined. The transmembrane regions were identified utilizing TMHMM2 (available at www.cbs.dtu.dk/services/TMHMM/) and SOSUI software v1 tools (accessible at http://www.cbs.dtu.dk and http://harrier.nagahama-i-bio.ac.jp) (accessed on 15 August 2023). The predicted transmembrane domains are indicated by purple peaks, while the central hydrophobic loop of proteins is represented by a blue line; Figure S6: The distribution of Gene Ontology terms within the OsAAP gene family was predicted using AgriGO. A. Biological Process. B. Cellular component. C. Molecular function; Figure S7: Quantile Q-Q plot for the culm number of *OsAAP17* genetic variants using linear regression analysis; Figure S8: Variance Importance plot analysis using random forest classifier of *OsAAP17* genetic variants; Figure S9: Genotyping of

rice varieties: (a), aus genotypes (b and c) and parents with population (d) using *OsAAP17-5* In/Del marker. Arrows represents the amplicon size. V—Rice varieties; BA-Aus genotypes; Figure S10: Phylogenetic analysis of aus lines and varieties via UPGMA model using in/dels allelic variant of *OsAAP17* gene. V—Released rice varieties, BA—BAAP Population, aus lines; Table S1: Genomic, CDS, protein, and promoter sequences of OsAAP; Table S2: A phylogenetic tree was constructed using AAP proteins from *O.sativa*, *A.thaliana*, *S. bicolor*, *S. moellendorffii*, and *P. patens*; Table S3: The distribution of duplicated rice AAP genes and the Ka/Ks ratio; Table S4: Putative motifs identified from OsAAP proteins using MEME. The sequence logos were generated using WebLogo; Table S5: In the promoter region of the OsAAP gene, there exists cis-regulatory elements; Table S6 AgriGo analysis has predicted a significant Gene Ontology (GO) term in the OsAAP gene family; Table S7: Annotation of the OsAAP gene via the use of eggNOGmapper; Table S8: Identified potential miRNAs and their AAP specific target genes in rice; Table S9: The network illustrating protein–protein interactions between OsAAP and other proteins in indica rice; Table S10: In/DelPrimer sequences of OsAAP17 gene; Table S11: a, Rice varieties used for the genetic diversity analysis. b, Aus species name used for genotyping of OsAAP17 allele; and Table S12: Mean weather data for both the growing seasons.

**Author Contributions:** C.P. (Chidambaranathan Parameswaran) and M.S.K. designed and wrote the manuscript; C.P. (Chidambaranathan Parameswaran) acquired funding; C.P. (Chidambaranathan Parameswaran) and M.S.K. supervised the study; the genotyping, data analysis, and preliminary draft preparation were conducted done by I.N. and B.S., the manuscript editing and finalization of the manuscript were performed by C.P. (Chinmay Pradhan) and C.B. The statistical analysis was completed by S.R.P. and J.L.K. Field experiments was coordinated by J.M. and overall coordination of the study was performed by S.S., S.-M.C., B.S.K. and A.-R.Z.G. M.S.H. provided valuable feedback to this study. All authors have read and agreed to the published version of the manuscript.

**Funding:** The assistance for this study has been provided through a general institute grant from the Indian Council of Agricultural Research (ICAR), New Delhi. Additionally, the authors wish to express their heartfelt gratitude to the Researchers Supporting Project number (RSPD2023R686) at King Saud University, Riyadh, Saudi Arabia.

**Institutional Review Board Statement:** Not applicable.

**Informed Consent Statement:** Not applicable.

**Data Availability Statement:** The data are available in the manuscript and in the Supplementary Materials.

**Acknowledgments:** The authors wish to express their heartfelt gratitude to the Researchers Supporting Project number (RSPD2023R686) at King Saud University, Riyadh, Saudi Arabia.

**Conflicts of Interest:** The authors declare no conflict of interest.

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
