# Peer review of "Genome-Wide Analysis of Amino Acid Transporter Gene Family Revealed That the Allele Unique to the Aus Variety Is Associated with Amino Acid Permease 17 (OsAAP17) Amplifies Both the Tiller Count and Yield in Indica Rice (Oryza sativa L.)"

_agronomy, doi:10.3390/agronomy13102629_

Round 1

Reviewer 1 Report

The article entitled Genome-wide analysis of the amino acid transporter gene family revealed that the allele unique to the Aus cultivar is associated with amino acid permease 17 (OsAAP17) and increases both tiller number and yield of indica rice (Oryza sativa L.) was reviewed and the following comments were made. The main objective of this study was to identify and characterize the AAP family in indica rice. In this study, a comprehensive genome-wide survey of the AAP gene family in rice was conducted using a number of computational tools.

The work is well presented and can be improved in structure. The manuscript is quite technical and the results are relevant to the field. The paper has a good number of citations, however only 24% are recent publications (within the last 5 years) and the number of self-citations is low. The results of the manuscript are reproducible according to the information in the methods section, but the document has the following shortcomings: It is too complex and contains so much information that the article is difficult to read and understand and ends up looking like a review article. The length and abundance of the charts can make it difficult for readers to read and understand the article, especially if they are not familiar with the topic or if the charts are not well explained in the text. This can limit the accessibility and usefulness of the article. It is important to ensure that each graphic included in the article is relevant to the research and contributes to the overall argument. Unnecessary graphics can overwhelm the reader and distract from the main message

Recommended:

- Summarize the article by highlighting the major components of the research paper.

- Better organize the "Materials" and "Methods" sections by placing them before the results and discussion.

- The conclusion of the article is filled with a lot of information that is not relevant to the discussion of the results.

 In general, the article is very relevant and contains valuable, accurate, technical and useful information. The proposed changes will allow it to reach more readers, to expand the form of dissemination in the scientific community.

Minor editing of English language required

Author Response

Reviewer 1

The article entitled Genome-wide analysis of the amino acid transporter gene family revealed that the allele unique to the Aus cultivar is associated with amino acid permease 17 (OsAAP17) and increases both tiller number and yield of indica rice (Oryza sativa L.) was reviewed and the following comments were made. The main objective of this study was to identify and characterize the AAP family in indica rice. In this study, a comprehensive genome-wide survey of the AAP gene family in rice was conducted using a number of computational tools. The work is well presented and can be improved in structure. The manuscript is quite technical and the results are relevant to the field. The paper has a good number of citations, however only 24% are recent publications (within the last 5 years) and the number of self-citations is low. The results of the manuscript are reproducible according to the information in the methods section, but the document has the following shortcomings: It is too complex and contains so much information that the article is difficult to read and understand and ends up looking like a review article. The length and abundance of the charts can make it difficult for readers to read and understand the article, especially if they are not familiar with the topic or if the charts are not well explained in the text. This can limit the accessibility and usefulness of the article. It is important to ensure that each graphic included in the article is relevant to the research and contributes to the overall argument. Unnecessary graphics can overwhelm the reader and distract from the main message.

Response: We sincerely appreciate your kind and thoughtful feedback on our manuscript. Your positive and constructive comments have been of great value to us, aiding us in the enhancement, refinement and improve the quality of our paper. We have meticulously reviewed your comments and have incorporated the necessary changes, marked in red using tracked changes. We trust that these revisions align with your expectations. Additionally, we have provided point-by-point responses to your comments below.

Minor comments

Comment 1: Summarize the article by highlighting the major components of the research paper? 

Response: Thank you for your valuable suggestion. Your input is greatly appreciated. We have highlighted the major component of the research in abstract and conclusion section and marked in red color in the manuscript.

Comment 2: Better organize the "Materials" and "Methods" sections by placing them before the results and discussion?

Response: We greatly appreciate your valuable suggestion. We have incorporated the Materials and Methods section before the results and discussion, and these sections have been highlighted using red color markers in the manuscript.

Comment 3: The conclusion of the article is filled with a lot of information that is not relevant to the discussion of the results. 

Response: Thank you for your valuable suggestions. In accordance with your advice, the conclusion section has been revised and these additions have been highlighted in red within the manuscript.

Comment 4: In general, the article is very relevant and contains valuable, accurate, technical and useful information. The proposed changes will allow it to reach more readers, to expand the form of dissemination in the scientific community.

Response: We sincerely appreciate your kind and thoughtful feedback on our manuscript.

Reviewer 2 Report

1.       This manuscript investigated amino acid transporter gene family in indica rice, and identified 27 OsAAP genes unevenly distributed across the 12 rice chromosomes. The attributes including classification, phylogenetic relationships, transcriptome profiling, miRNAs with a specific affinity for OsAAP genes, etc, of these OsAAP genes were delved. Interestingly, the authors claimed that a aus unique allele of OsAAP3(OsAAP17) was associated with culm  number in indica rice, which was valuable in genetic improvement.

2.       As the agronomical traits especially tiller number were sensitive to the cultivation environments, the phenotype evaluation from a recombinant inbred population created by crossing N22 and JR201 in two seasons cannot obtain consistent results, therefore, the authors concluded that the increasement of tiller number induced by N22-allele was more pronounced during the dry cultivation season, indicating the role of environmental factors in OsAAP17-mediated regulation of tiller numbers. However, this result needs to be validated in further experience of two Seasons (Kharif and Rabi) environments to support the authors’ conclusion.

3.       In lines 216-219, 27 OsAAP genes was categorized into 16 distinct subfamilies. In some subfamilies, i.e. 4,5,9,12, there is not any member in them, how can them as a group or subfamily? If the classification of OsAAP genes was referred from Figure 1, the number of OsAAP genes in each subgroup was not consistent to the number of corresponding group of figure 1.

4.       In figure 2, the section (B) is no need as the information was presented in section (A).

5.       In lines 346, 350, please supply the annotation of the abbreviation of “ARE”, and “MBS”.

6.       In table 3, please supply the annotation of the abbreviation of “S1. No”.

7.       In lines 499-508, please supply explanation of the results to the readers how the result implys.

8.       In section 2.10, please supply accurate define of the “131 different rice varieties” and the “24 aus genotypes”, as in general concept, aus varieties also included in “different varieties”.

9.       In table 5, the numerical values of some parameters such as Genotype No, Sample Size, No. of obs., No. of obs., should be expressed as integers, without the need for decimal points or subsequent zero symbols.

10.   In table 6, please supply detail information of the mapping population, such as population type, generation(F2? Fn? BCnFn?), sample size, etc.

11.   Please check the value of table 6, in line 546, the statement of “Furthermore, single plant yields for seasons I and II were 17.14 and 16.82 g, respectively.” is not consistent to the value in table 6.

12.   In table 6a, please supply a uniform name to “OsAAP17” or “AAP3”, and it is need to be expressed in a uniformative name in full text. In this table, “480 bp”, and “500bp” could be replaced by “JR201-allele” and “N22-allele”.

13.   Please check figure 11 carefully. For instance, in figure A, b, and c, percent change in the tiller number value of N22 allele of AAP3 over JR201 is 6.66 (%?) in season I, and 22.55 (%?) in season II, but the mean value is -16.87(%?), it is not reasonable. The value of other traits should be carefully checked.

Author Response

Reviewer 2

This manuscript investigated amino acid transporter gene family in indica rice, and identified 27 OsAAP genes unevenly distributed across the 12 rice chromosomes. The attributes including classification, phylogenetic relationships, transcriptome profiling, miRNAs with a specific affinity for OsAAP genes, etc, of these OsAAP genes were delved. Interestingly, the authors claimed that a aus unique allele of OsAAP3 (OsAAP17) was associated with culm  number in indica rice, which was valuable in genetic improvement.

Response: We sincerely appreciate your kind and thoughtful feedback on our manuscript. Your positive and constructive comments have been of great value to us, aiding us in the enhancement, refinement and improve the quality of our paper. We have meticulously reviewed your comments and have incorporated the necessary changes, marked in red using tracked changes. We trust that these revisions align with your expectations. Additionally, we have provided point-by-point responses to your comments below.

Major Comments:

      Comment 1: As the agronomical traits especially tiller number were sensitive to the cultivation environments, the phenotype evaluation from a recombinant inbred population created by crossing N22 and JR201 in two seasons cannot obtain consistent results, therefore, the authors concluded that the increasement of tiller number induced by N22-allele was more pronounced during the dry cultivation season, indicating the role of environmental factors in OsAAP17-mediated regulation of tiller numbers. However, this result needs to be validated in further experience of two Seasons (Kharif and Rabi) environments to support the authors’ conclusion.

      Response: Thank you for your valuable feedback. We also found it intriguing that the environmental impact of the aus allele of OsAAP17 was more pronounced during dry cultivation seasons. Additionally, our classification analysis and genetic diversity analysis revealed that aus genotypes possess a unique genetic makeup for the OsAAP17 (AAP3) gene in rice. Moreover, the aus allele of OsAAP17 exhibited consistent tiller numbers in both seasons (Season I: 10.5; Season II: 9.8), while the indica allele displayed variations (Season I: 9.8; Season II: 7.59). This suggests the stability of the aus allele in influencing tiller numbers, in contrast to the indica allele, which appears to be more influenced by environmental factors. Given that aus genotypes are known for their stability across various environmental conditions, this discovery holds significant promise for rice improvement programs. We plan to further investigate this effect on a larger scale in future experiments. Therefore, we kindly request that this information be taken into consideration.

     Comment 2: In lines 216-219, 27 OsAAP genes was categorized into 16 distinct subfamilies. In some subfamilies, i.e. 4,5,9,12, there is not any member in them, how can them as a group or subfamily? If the classification of OsAAP genes was referred from Figure 1, the number of OsAAP genes in each subgroup was not consistent to the number of corresponding group of figure 1.

Response: Thanks a lot for constructive feedback. To construct the phylogenetic tree for this analysis, MEGA 11 was employed, incorporating 27 AAP proteins from O. sativa, 8 AAP proteins from A. thaliana, 21 AAP proteins from S. bicolor, 8 AAP proteins from S. moellendorffii, and 5 AAP proteins from P. patens (Figure 1, Table S2). The outcomes demonstrated that the 69 AAP genes from the model and other plant species could be categorized into 16 distinct subfamilies. Among these, Group XV comprised 8 members, while Groups I, II, III, IV, V, VI, VII, VIII, IX, X, XI, XII, XIII, XIV, and XVI contained 2, 2, 1, 0, 0, 1, 1, 1, 0, 1, 2, 0, 1, 4, and 3 members, respectively (Figure S2). We corrected it and mark in red color in the manuscript.

    Comment 3: In figure 2, the section (B) is no need as the information was presented in section (A).

    Response: We have addressed the issue as per the reviewer's suggestions. We have removed Figure 2B and the corresponding red-colored alterations within the manuscript.

    Comment 4: In lines 346, 350, please supply the annotation of the abbreviation of “ARE”, and “MBS”.

    Response: Thanks, as per reviewer's suggestions, we have included the full forms of ARE (Adenine-rich Element), LTR (low-temperature responsiveness), and MBS (MYB-binding Site) throughout the manuscript and mark in red color in the manuscript.

    Comment 5: In table 3, please supply the annotation of the abbreviation of “S1. No”. 

    Response: Thanks for your valuable suggestion. we have included the full forms of Sl. No. (Serial number) and mark in red color in the manuscript.

    Comment 6: In lines 499-508, please supply explanation of the results to the readers how the result implys.

Response: Thanks for your valuable suggestion. In response to the reviewer's suggestions, we have provided detailed explanations of the results to enhance reader understanding. These explanatory sections have been highlighted in red within the manuscript. The identified protein-protein interactions can contribute to a more profound comprehension of the intricate regulatory mechanisms involved in both biotic and abiotic stress responses, as well as the roles of OsAAP gene family members in growth and development processes. These findings present valuable insights that can propel further investigations into the functional characterization of OsAAP genes, shedding light on their roles and activities. Thus, this knowledge can ultimately aid in the development of high-yielding and stress-tolerant rice cultivars.

Comment 7: In section 2.10, please supply accurate define of the “131 different rice varieties” and the “24 aus genotypes”, as in general concept, aus varieties also included in “different varieties”

Response: We appreciate your valuable suggestion. The manuscript has been updated accordingly in sections 2.10, 3.6, and 4.7 to reflect that the 131 rice varieties are of the indica type and are commercially cultivated, while the 24 aus genotypes are typically landraces not commonly grown commercially by farmers.

Comment 8: In table 5, the numerical values of some parameters such as Genotype No, Sample Size, No. of obs., No. of obs., should be expressed as integers, without the need for decimal points or subsequent zero symbols. 

Response: Thank you for your feedback. The suggested changes have been implemented in the table 5.

Comment 9: In table 6, please supply detail information of the mapping population, such as population type, generation(F2? Fn? BCnFn?), sample size, etc.

Response: Thanks, the study involves a population of F6 RILs (Recombinant Inbred Lines) with a sample size of 48, comprising 24 individuals with the aus allele of OsAAP17 and 24 individuals with the indica allele of OsAAP17. Additionally, two parent lines, N22 and JR201, are included in the analysis. We updated in the manuscript and marked in red color in the manuscript.

    Comment 10: Please check the value of table 6, in line 546, the statement of “Furthermore, single plant yields for seasons I and II were 17.14 and 16.82 g, respectively.” is not consistent to the value in table 6.

Response: Thank you very much for your feedback. The error was unintentional, and it has been rectified in the revised manuscript. The corrected statement now reads: "Furthermore, single plant yields for seasons I and II were 17.14 and 15.97 g, respectively and mark in red color in the manuscript.

Comment 11: In table 6a, please supply a uniform name to “OsAAP17” or “AAP3”, and it is need to be expressed in a uniformative name in full text. In this table, “480 bp”, and “500bp” could be replaced by “JR201-allele” and “N22-allele”.

Response: Thanks a lot for the comments. The name of the gene is made uniform as OsAAP17. The allele named is changed as JR201 allele (480 bp) and N22 allele (500 bp) in the revised manuscript and mark in red color in the manuscript.

Comment 12: Please check figure 11 carefully. For instance, in figure A, b, and c, percent change in the tiller number value of N22 allele of AAP3 over JR201 is 6.66 (%?) in season I, and 22.55 (%?) in season II, but the mean value is -16.87(%?), it is not reasonable. The value of other traits should be carefully checked. 

Response: We appreciate your valuable input. In Figure 11 A and B, a comparison was conducted between the JR201 allele and N22 allele for various traits. For instance, in Figure 11A, during Season I, the mean tiller number for the JR201 allele was 9.8, while for the N22 allele, it was 10.5. This resulted in a percent difference of 6.66%, indicating that the N22 allele had a slightly higher tiller number in Season I. Similarly, in Figure 11B, the mean tiller number for the JR201 allele was 7.59, whereas for the N22 allele, it was 9.8. This yielded a percent difference of 22.5%, indicating a substantial difference in tiller numbers between the two alleles in Season II. However, Figure 11C represents a comparison across all RILs (not categorized by N22 and JR201 alleles of AAP17) in both seasons. In Season I, the mean tiller number was 10.18, whereas in Season II, it decreased to 8.71, resulting in a percent difference of -16.87%. This suggests that, when considering the entire population, there was a reduction in tiller numbers from Season I to Season II. Interestingly, when looking at specific lines with the N22 allele of AAP17, they showed a relatively consistent tiller number between Season I and Season II (10.5 and 9.8, respectively). The reduction in tiller number in Season II was primarily observed in RILs carrying the JR201 allele of OsAAP17. We have, therefore, revised the legend for Figure 11.C to read as follows: 'The percentage difference in mean yield-related traits between Season II compared to Season I within the mapping population for both seasons.'" Notably, the modifications have been highlighted in the revised manuscript using red text color.

Reviewer 3 Report

Genome-wide Analysis of Amino Acid Transporter Gene 2 Family Revealed that the Allele Unique to the Aus Variety is 3 Associated with Amino Acid Permease 17 (OsAAP17), Amplifies 4 both Tiller Count and Yield in Indica Rice (Oryza sativa L.)

In this study, complex identification revealed a total of 27 AAP family genes in the rice indica genome. Subsequently, the genes were classified into sixteen separate subfamilies, the classification was achieved by assessing their sequential composition and phylogenetic relationships. Analysis of cis elements in OsAAP genes showed that OsAAP promoters include phytohormone, plant growth and development, as well as a cis element associated with stress. Thus, the identified allelic variations in the use of OsAAP17 can potentially increase the yield of rice by molecular selection.

The topic of the work is relevant. The design of the study corresponds to its goals and objectives, and the conclusions are supported by the evidence presented. The methods are described enough to be able to repeat the study. The use of statistics and data processing is appropriate. The form of submission of the work is clear. The images in this manuscript are free from obvious manipulation.

 The section "Material and Methodology" should be placed before the section "Results". 

Author Response

Reviewer 3

Genome-wide Analysis of Amino Acid Transporter Gene 2 Family Revealed that the Allele Unique to the Aus Variety is 3 Associated with Amino Acid Permease 17 (OsAAP17), Amplifies 4 both Tiller Count and Yield in Indica Rice (Oryza sativa L.) In this study, complex identification revealed a total of 27 AAP family genes in the rice indica genome. Subsequently, the genes were classified into sixteen separate subfamilies, the classification was achieved by assessing their sequential composition and phylogenetic relationships. Analysis of cis elements in OsAAP genes showed that OsAAP promoters include phytohormone, plant growth and development, as well as a cis element associated with stress. Thus, the identified allelic variations in the use of OsAAP17 can potentially increase the yield of rice by molecular selection. The topic of the work is relevant. The design of the study corresponds to its goals and objectives, and the conclusions are supported by the evidence presented. The methods are described enough to be able to repeat the study. The use of statistics and data processing is appropriate. The form of submission of the work is clear. The images in this manuscript are free from obvious manipulation

Response: We sincerely appreciate your kind and thoughtful feedback on our manuscript. Your positive and constructive comments have been of great value to us, aiding us in the enhancement, refinement and improve the quality of our paper. We have meticulously reviewed your comments and have incorporated the necessary changes, marked in red using tracked changes. We trust that these revisions align with your expectations. Additionally, we have provided point-by-point responses to your comments below.

Minor Comments:

Comment 1: The section "Material and Methodology" should be placed before the section "Results".

Response: We greatly appreciate your valuable suggestion. We have incorporated the Materials and Methods section before the results and discussion, and these sections have been highlighted using red color markers in the manuscript.

Round 2

Reviewer 2 Report

1. Tiller number is a quantitative trait which is sensitive to the environments, the tiller number of aus allele and indica allele in OsAAP17 showed no significant difference in the first season, but apparent difference in the second season, therefore, the authors claimed that aus allele performance stable tiller number in drought environment, the target aus allele of OsAAP 17 showed potential effect in increasing tiller number; it is interesting for breeding programs. However, as a quantitative trait, the phenotype needs to be confirmed in more environments (one more Kharif and one more Rabi) to analysis genotype x environment interaction effect, and avoid false conclusion. Unfortunately, the authors did not supply further information to support their conclusion.

2. Some statements in the text were still not consistent to experimental data, for instance, in lines 367-371, the statement is not consistent to the figure 1. In lines 713-715, in season II, the mean filled and unfilled grain is 63.12 and 46.66 is not consistent to that in table 6 (82.33, 27.45 respectively).

Author Response

Reviewer 2

This manuscript investigated amino acid transporter gene family in indica rice, and identified 27 OsAAP genes unevenly distributed across the 12 rice chromosomes. The attributes including classification, phylogenetic relationships, transcriptome profiling, miRNAs with a specific affinity for OsAAP genes, etc, of these OsAAP genes were delved. Interestingly, the authors claimed that a aus unique allele of OsAAP3 (OsAAP17) was associated with culm  number in indica rice, which was valuable in genetic improvement.

Response: We sincerely appreciate your kind and thoughtful feedback on our manuscript. Your positive and constructive comments have been of great value to us, aiding us in the enhancement, refinement and improve the quality of our paper. We have meticulously reviewed your comments and have incorporated the necessary changes, marked in red using tracked changes. We trust that these revisions align with your expectations. Additionally, we have provided point-by-point responses to your comments below.

Major Comments:

      Comment 1: As the agronomical traits especially tiller number were sensitive to the cultivation environments, the phenotype evaluation from a recombinant inbred population created by crossing N22 and JR201 in two seasons cannot obtain consistent results, therefore, the authors concluded that the increasement of tiller number induced by N22-allele was more pronounced during the dry cultivation season, indicating the role of environmental factors in OsAAP17-mediated regulation of tiller numbers. However, this result needs to be validated in further experience of two Seasons (Kharif and Rabi) environments to support the authors’ conclusion.

      Response: We appreciate your comment. The Aus allele of OsAAP17 displayed uniqueness as indicated in Table 4. To validate these results, we initiated the development of a mapping population in the 2016-17 season. Subsequently, we evaluated the F6 lines for traits related to yield in two distinct seasons, namely Kharif 2021 and Rabi 2022. Furthermore, the observed differences were statistically significant. Our future plans include conducting expression analysis and comprehensive yield evaluations on a larger scale, using plots spanning an area of 10 square meters

     Comment 2: Some statements in the text were still not consistent to experimental data, for instance, in lines 367-371, the statement is not consistent to the figure 1.

Response: Thanks a lot for constructive feedback. To construct the phylogenetic tree for this analysis, MEGA 11 was employed, incorporating 27 AAP proteins from O. sativa, 8 AAP proteins from A. thaliana, 21 AAP proteins from S. bicolor, 8 AAP proteins from S. moellendorffii, and 5 AAP proteins from P. patens (Figure 1, Table S2). The phylogenetic analysis revealed that the 69 AAP proteins from the model and other plant species could be categorized into 16 distinct subfamilies. However, the OsAAP gene family members belong to the Group XV which is comprised highest 8 members of OsAAP gene family, while Groups I, II, III, IV, V, VI, VII, VIII, IX, X, XI, XII, XIII, XIV, and XVI contained 2, 0, 1, 0, 0, 1, 1, 2, 0, 1, 2, 0, 1, 4, and 3 members, respectively (Figure S2). We corrected it and mark in red color in the manuscript.

Figure S2. Distribution of OsAAPs within a distinct cluster of the phylogenetic tree.

     Comment 3: In lines 713-715, in season II, the mean filled and unfilled grain is 63.12 and 46.66 is not consistent to that in table 6 (82.33, 27.45 respectively).

    Response: Thank you for your positive response. The suggested changes have been implemented and mark in red color in the manuscript.

Round 3

Reviewer 2 Report

The comments are based on the file “agronomy-2617697-track change”:

1.       Lines 744-758, in season I, the mean difference for different traits for both OsAAP17 alleles in the mapping population was found to be non-significant. The mean difference for tiller number between OsAAP17 alleles was significant only in season II. So, in lines 54-56, the statement “Moreover, the aus allele of OsAAP17 increased the number of productive tillers and single plant yield by 22.7% and 9.6%, respectively, in a recombinant inbred population created by crossing N22 and JR 201.” should be revised to indicate this result is only from season II, because the cited data produced from season II.  

2.       Authors should carefully examine the data in full text. For instance, lines 750-752, the statement of “7.59 and 9.83, respectively” should be check correponding table 6B (7.59 and 9.8, respectively), and the difference could be 2.21, but not 2.23?

3.       Line 52, “rice varieties” could be instead by “indica rice varieties”?

4.       Line 55, the data “22.7% and 9.6%” could be revised to the corresponding after the revision of lines 750-752?

5.       Lines 253, 277, “Aus” to “aus”; line 328, ““Indica” to “indica””. In table 4, “Aro” to “aro”, “Indica” to “indica”……, please check it carefully in all text.

6.       In lines 361-362, the statement of “the OsAAP gene family members belong 361 to the Group XV……” could be consider to be revised, or it may confuse.

Author Response

Ms. Carrie Tu,
Assistant Editor,
Agronomy Editorial Office, MDPI,

Haidian, Beijing, China

Manuscript ID: agronomy-2617697

Subject: Submission of the revised manuscript in “Agronomy"

Dear Chief in Editor and Editorial Team,

We extend our gratitude to you for taking the time to consider the manuscript (ID: agronomy-2617697) for revision. Substantial improvements have been made to the manuscript in accordance with the feedback from the reviewers. We appreciate the valuable insights provided by the reviewers, which have greatly contributed to enhancing the quality of the manuscript. Our detailed responses to each reviewer's comments are provided below.

Sincerely, with best regards

Mahipal Singh Kesawat P.hD.

Assistant Professor,

Faculty of Agriculture,  

Sri Sri University,

Cuttack, Odisha, India

Mobile No: +91-8306808940  

Reviewer 2

This manuscript investigated amino acid transporter gene family in indica rice, and identified 27 OsAAP genes unevenly distributed across the 12 rice chromosomes. The attributes including classification, phylogenetic relationships, transcriptome profiling, miRNAs with a specific affinity for OsAAP genes, etc, of these OsAAP genes were delved. Interestingly, the authors claimed that a aus unique allele of OsAAP3 (OsAAP17) was associated with culm  number in indica rice, which was valuable in genetic improvement.

Response: We sincerely appreciate your kind and thoughtful feedback on our manuscript. Your positive and constructive comments have been of great value to us, aiding us in the enhancement, refinement and improve the quality of our paper. We have meticulously reviewed your comments and have incorporated the necessary changes, marked in red using tracked changes. We trust that these revisions align with your expectations. Additionally, we have provided point-by-point responses to your comments below.

Major Comments:

      Comment 1: Lines 744-758, in season I, the mean difference for different traits for both OsAAP17 alleles in the mapping population was found to be non-significant. The mean difference for tiller number between OsAAP17 alleles was significant only in season II. So, in lines 54-56, the statement “Moreover, the aus allele of OsAAP17 increased the number of productive tillers and single plant yield by 22.7% and 9.6%, respectively, in a recombinant inbred population created by crossing N22 and JR 201.” should be revised to indicate this result is only from season II, because the cited data produced from season II.

      Response: Thank you for the comment. The rabi season was mentioned in the abstract and the sentence was reframed as ‘Moreover, in the Season II (rabi season) it was found that the aus allele of OsAAP17 increased the number of productive tillers and single plant yield by 22.55% and 9.67%, respectively, in a recombinant inbred population created by crossing N22 and JR 201”.

     Comment 2: Authors should carefully examine the data in full text. For instance, lines 750-752, the statement of “7.59 and 9.83, respectively” should be check corresponding table 6B (7.59 and 9.8, respectively), and the difference could be 2.21, but not 2.23?.

Response: Thank you for the comment. The data has been carefully rechecked, and we have made the necessary changes accordingly.

    Comment 3: Line 52, “rice varieties” could be instead by “indica rice varieties”?.

    Response: Thank you for your feedback. We have included the 'indica rice varieties' as you suggested, and these additions have been marked in red within the manuscript.

    Comment 4: Line 55, the data “22.7% and 9.6%” could be revised to the corresponding after the revision of lines 750-752?

    Response: Thank you for your comment. We have updated the values to 22.55% and 9.67% in both the abstract and results section, and you can find these changes highlighted in red in the manuscript.

    Comment 5: Lines 253, 277, “Aus” to “aus”; line 328, ““Indica” to “indica””. In table 4, “Aro” to “aro”, “Indica” to “indica”……, please check it carefully in all text.

    Response: Thank you for your feedback. We have made the recommended changes in the manuscript, and you can find these revisions highlighted in red.

    Comment 6: In lines 361-362, the statement of “the OsAAP gene family members belong 361 to the Group XV……” could be consider to be revised, or it may confuse.

    Response: Thank you for your feedback. In the manuscript, we have revised the sentence, and you can find the updated version highlighted in red.
